# Glacial dysoxia in the deep subpolar North Atlantic during the Mid-Pleistocene Transition

Iván Hernández-Almeida [1,2] ✉, Francisco Javier Sierro [3], Gabriel M. Filippelli [4], Antje H. L. Voelker [5,6] & Paula Diz [7]

The transition from 41-kyr to 100-kyr climate cycles during the Mid-Pleistocene Transition (MPT; 1250–700 kyr), occurred in the absence of any significant shift in orbital forcing. The increase of carbon storage in the deep ocean between Marine Isotope Stage (MIS) 24–22 has been suggested as a main internal factor leading to the transition to 100-kyr glacial cycles. We present sedimentary redox proxies and benthic foraminifera assemblages that demonstrate persistent dysoxic conditions during the MPT at IODP Site U1314 (subpolar North Atlantic). During glacials between 940 and 870 kyr, benthic foraminifera species typical of high porewater oxygen concentration disappeared, and concentrations of manganese oxides and reactive phosphorus mineral phases, both influenced by redox state, showed reductions exceeding 50%. Here, we show that higher freshwater supply associated with ice-rafted delivery caused a reduction in deep-water convection, decreasing bottom-water oxygenation and favoring carbon storage in the subpolar North Atlantic during the MPT.

During the Mid-Pleistocene Transition (MPT), high-amplitude 100-kyr cycles emerged in surface and deep ocean climate records, despite the absence of concurrent changes in external orbital forcing[1,2]. Atmospheric $p$CO$_2$ reconstructions indicate a 30 ppm decrease between 1,000 and 800 kyr ago[3–5]. The deep ocean, Earth's largest carbon reservoir that readily exchanges with the atmosphere, has been proposed to play a key role in carbon sequestration[6–8], and could mitigate the climate change risks of global warming[9].

The leading hypothesis suggests that the initiation of 100-kyr glacial cycles during the MPT was driven by increased carbon storage in the deep ocean[6,8,10,11]. The Southern Ocean has emerged as the leading region to explain the climate feedbacks during the MPT. Potential and not mutually exclusive mechanisms include enhanced iron fertilization and the biological pump[12,13], increase ice-sheet and sea-ice extent[7,14,15], stronger interglacial Antarctic circumpolar circulation[16], strengthening of the halocline[17,18] and isolation of Antarctic bottom waters[8,17,19,20]. These studies agree that the glacial expansion of Southern Ocean sourced waters would provide a deep ocean reservoir for respired CO$_2$ during the MPT[8,11,19–22]. An enhanced glacial marine carbon sequestration would also cause reduced oxygen in the ocean interior. A significant decrease in bottom-pore-water oxygenation (BPWO) is documented in some parts of the mid-latitude deep Atlantic[10,23] and Pacific[24,25], particularly between ~940-870 kyr. Widespread incursions of southern source waters (equivalent to modern Antarctic Bottom Water -AABW-) during glacial periods would require a reconfiguration of the glacial Atlantic meridional overturning circulation (AMOC), which imply changes in the production mode and/or strength of North Atlantic Deep Water (NADW) formed in the high-latitude North Atlantic[6,26]. If NADW was substantially less

[1]Department of Earth and Planetary Science, Swiss Federal Institute of Technology Zurich, Zürich, Switzerland. [2]PAGES IPO Bern, Bern, Switzerland. [3]Department of Geology, University of Salamanca, Salamanca, Spain. [4]Department of Earth and Environmental Science, Indiana University, Indianapolis, IN, USA. [5]Instituto Português do Mar e da Atmosfera (IPMA), Divisão de Geologia e Georecursos Marinhos, Algés, Portugal. [6]Centro de Ciências do Mar (CCMAR/CIMAR LA), Universidade do Algarve, Faro, Portugal. [7]Centro de Investigación Mariña, XM1, Universidade de Vigo, Vigo, Spain. ✉e-mail: ivan.hernandez@unibe.ch

vigorous or shallower during glacial intervals across the MPT, a change in the BPWO near deep convection areas in the northern North Atlantic should be expected, which could potentially involve the build-up of a respired carbon reservoir also in the high-latitude Northern Hemisphere. However, absence of oxygen-sensitive records near deep-convection areas in the northern North Atlantic precludes testing this hypothesis.

Here we analyzed geochemical and micropaleontological proxies that suggest that dysoxic conditions[<5–10 µmol/kg $O_2$[27]; occurred during intervals with reduced NADW in the subpolar North Atlantic driven by meltwater forcing at glacial intervals during the MPT (1069-779 kyr). We present multiple independent proxies of bottom and porewater oxygen (BPWO) depletion (solid-phase manganese (Mn) enrichments preserved in bulk sediments and planktonic foraminifera coatings, sedimentary phosphorus (P) reservoirs, and benthic foraminifera species) from Integrated Ocean Drilling Program (IODP) Site U1314, located in the subpolar North Atlantic (Fig. 1A). In combination with previously published sedimentological proxies that document changes in freshwater supply to the ocean due to ice melting[28]), our results provide evidence of decreased BPWO in the subpolar North Atlantic related to changes in deep water production in that region. Moreover, comparison with paleoceanographic and redox-sensitive proxies from the Southern Ocean suggests that coupling between both regions was key in setting AMOC dynamics to facilitate increased carbon sequestration. Similarity of BPWO changes in both polar regions helps to explain the changes in deep ocean ventilation and carbon storage during the emergence of 100-kyr glacial cycles in the MPT.

## Results and discussion
### Records of past BPWO during the Mid-Pleistocene Transition
New and previously published BPWO proxy records from Sites U1314 and 1094, both located downstream of newly formed deep-water masses in both hemispheres, provide evidence of changes in deep-ocean oxygenation during the MPT. These sites have high-resolution and continuous sedimentation over the MPT, robust chronologies and available datasets of proxies of surface and deep oceanography (Supplementary Information; Supplementary Fig. 1)[17,29,30]. IODP Site U1314 (56.36°N, 27.82°W, water depth 2800) m is bathed today by the well oxygenated Iceland-Scotland overflow waters (ISOW; 294 µmol/kg), which are formed during wintertime convection in the Nordic Seas[31]. This water mass is a major source for the AMOC lower limb, the NADW[32] (Fig. 1A, B). Surface hydrography at Site U1314 is controlled by the seasonal shift of the subarctic front (SAF), which separates southward flowing cold polar waters with occasional sea-ice formation from the northward flowing warm Atlantic waters[33]. Shifting of the SAF south of Site U1314 led to cold conditions and seasonal sea-ice formation, and episodic ice-rafted debris (IRD) discharges during the MPT[28,29]. In particular, the lower benthic foraminifera carbon isotope ($\delta^{13}C_b$) values at Site U1314 (Fig. 2A)[29] and at other sites in the Atlantic basin during MIS 24–22 (Supplementary Fig. 2) may reflect a lower export of NADW or the intrusion of southern-source waters with higher nutrient concentrations[6,7]. These observations suggest strong coupling between surface hydrographic conditions and ventilation of the deep Atlantic. Ocean Drilling Program (ODP) Site 1094 (53.2°S, 5.1°E, water depth 2807 m) is in the South Atlantic Ocean, south of the Antarctic polar front (APF) and close to the average winter sea-ice edge. This site is currently well ventilated by newly formed AABW (dissolved $O_2$: 217 µmol/kg)[34] (Fig. 1A, C). Late Pleistocene sediment

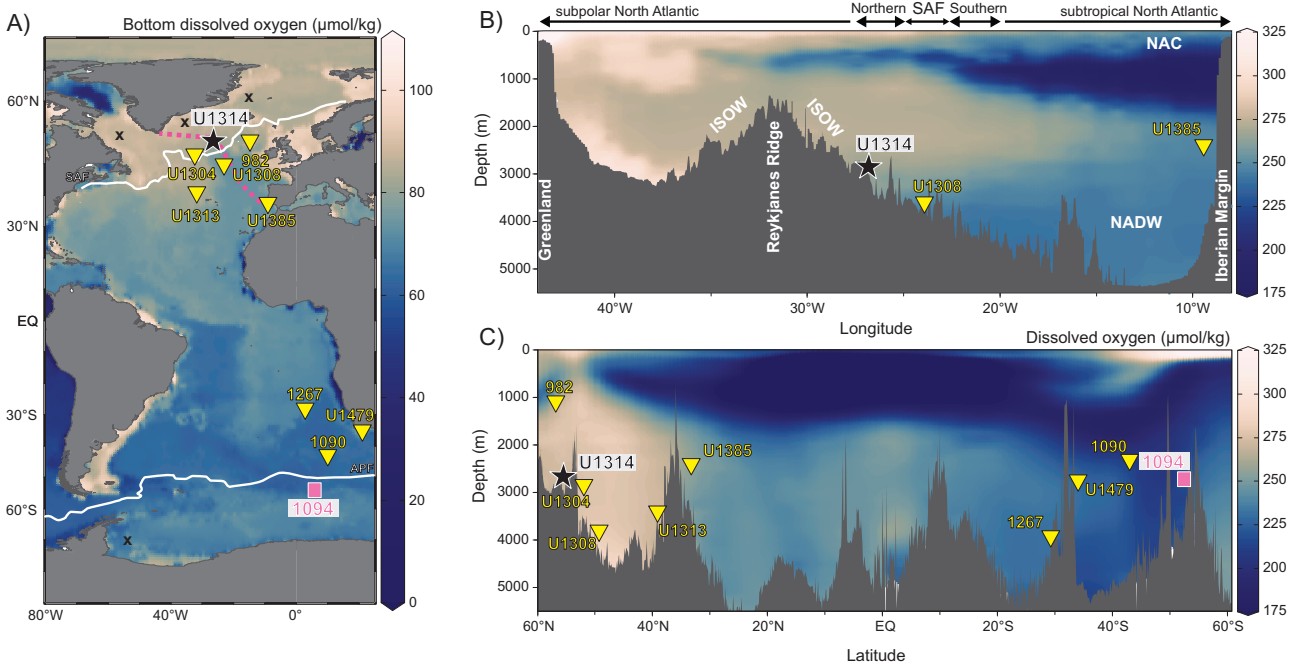

**Fig. 1 | Location of sediment cores and modern Atlantic Ocean bottom dissolved oxygen. A** IODP Site U1314 (black star) in the North Atlantic, along with other cores located in the Atlantic basin; Sites 982, U1304, U1308, U1313, U1385, 1267, 1090, U1479 (yellow triangles) and 1094 (pink square). The shaded colors show oxygen concentration at the sea floor from World Ocean Atlas 2018 data[84]. White lines indicate the position of the average geographical position of the modern SAF [which corresponds approximately to the 10 °C isotherm[85]; and APF today, and black "X" symbol marks regions of modern-day deep-water formation. Pink-dashed line marks the cross-section shown in panel 1B. **B** Cross-section showing modern-day oxygen concentrations (color scale) across the OVIDE/A25 transect[86]. SAF: subarctic front; ISOW: Iceland-Scotland Overflow Water; NADW: North Atlantic Deep Water; NAC: North Atlantic Current. **C** Meridional cross-section of the Atlantic basin showing modern-day oxygen concentrations (data from World Ocean Atlas 2018[84]). Oxygen-rich waters formed in the Nordic Seas overflow the Iceland-Scotland Ridge and fill depths >1.5 km depth. Northward flowing Antarctic intermediate waters with lower oxygen concentrations occupy the upper 1.5 km depth. Plots created using Ocean Data View[87].

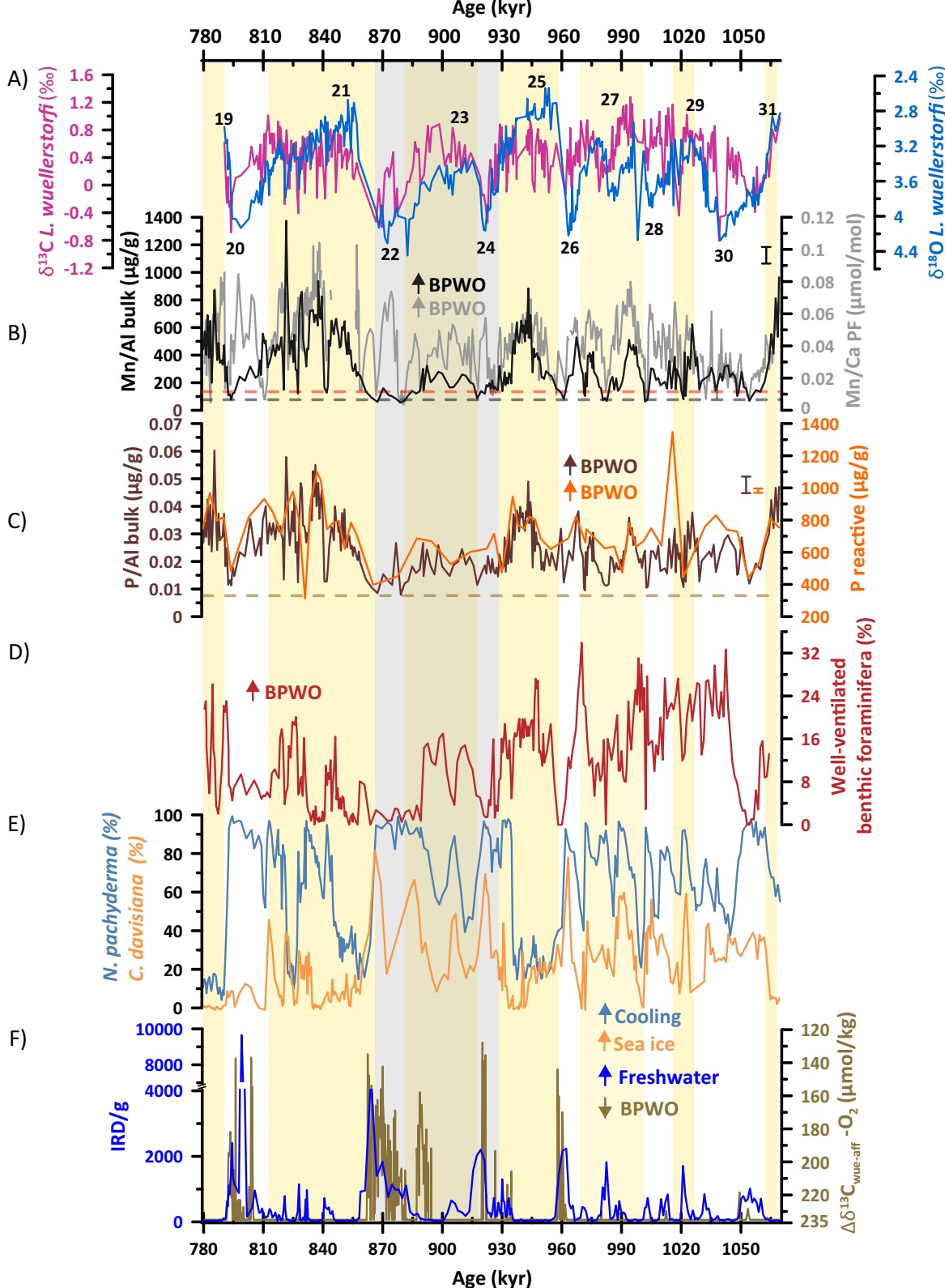

records from this site show sensitivity to changes in deep ocean oxygenation, and therefore ODP Site 1094 is a good location to track changes in the AABW[34]. Previously published Mn/Al ratios in bulk sediments and Mn/Ca ratios in planktonic foraminifera coatings at these two sites are used as proxies for porewater oxygenation (Supplementary Information). Under oxic sedimentary conditions, dissolved Mn in porewater precipitates as solid-phase Mn-

enrichments[35,36]. Reduction of $Mn^{4+}$ to soluble $Mn^{2+}$ occurs under low BPWO, leading to the dissolution of the precipitated Mn-rich minerals, and then $Mn^{2+}$ is released into solution in porewaters[35]. Upon return to oxidizing conditions, the dissolved $Mn^{2+}$ can re-precipitate in the sediment as a Mn-rich authigenic mineral (e.g., Mn oxides or carbonates)[35,37], where the foraminiferal calcite tests act as a nucleation template for secondary mineral precipitation[36]. At sites U1314 and

**Fig. 2 | Downcore isotopic, micropaleontological and geochemical records from IODP Site U1314. A** Stable isotope records of benthic foraminifera *Lobatula wuellerstorfi* (blue line δ[18]O and pink line δ[13]C vs. per mil VPDB)[29,30]. Black numbers indicate interglacials (odd) and glacials (even) marine isotope stages (MIS). **B** Mn/Ca ratio in planktonic foraminifera coatings (Mn/Ca PF, gray line)[72] and Mn/Al in bulk sediment (black line). Horizontal black dashed line indicates average upper continental crust compositions (i.e., 0.0075 for Mn/Al), and the red dashed line indicates average andesitic crust composition (i.e., 0.0116 for Mn/Al)[74]. **C** Bulk P/Al (brown line) and P reactive (includes P-Fe bound, organic, authigenic) from sequential extraction (orange line). Horizontal brown dashed line indicates average upper continental crust compositions (i.e., 0.007 P/Al)[74]. **D** Relative abundance of deep-sea species related to well-ventilated bottom-waters (*L. wuellerstorfi*,

*Gyroidina umbonata* and *Astronomion novozealandicum*). **E** Relative abundance (inverted axis) of polar and sea-ice related plankton species *Neogloboquadrina pachyderma* (light blue line) and *Cycladophora davisiana* (light orange line)[28,30], where high percentages of this species correspond to low BPWO at Site U1314. **F** IODP Site U1314 IRD/g (note that axis is truncated between 4000–7000 IRD/g)[29] (dark blue line) and IODP Site U1385 O₂ concentration based on carbon isotope gradients between *L. wuellerstorfi* and *Globobulimina affinis* (Δδ[13]C$_{wue-aff}$) (golden line, reversed axis). The O₂ reconstruction is truncated at 235 μmol kg⁻¹, above which the calibration is unreliable[23]. The vertical gray bar highlights MIS 24–MIS 22. Yellow vertical bars correspond to interglacial stages. Error bars are replicate 1 SD (Methods).

1094, Mn/Ca in planktonic foraminifera *Neogloboquadrina pachyderma* increases at the glacial-interglacial transitions. At U1314, these increases also occur during interglacial periods, indicative of more oxic conditions. Lower Mn/Ca values are observed when the oxygen penetration depth is shallower, mainly during glacial intervals. Positive correlations between Mn/Al in bulk sediments and Mn/Ca in planktonic foraminifera coatings during glacial/interglacial intervals are observed in these two sites (Supplementary Fig. 3A-C), indicating a common mechanism driving the precipitation of solid-phase Mn-enrichments related to the level of porewater oxygenation depending on the dissolved oxygen concentration of deep-water masses at Sites U1314 (NADW) and 1094 (AABW).

Total P and reactive P phases (organic, P-Fe bound and authigenic, Supplementary Fig. 4) in bulk sediments at Site U1314 also record redox conditions in the sediment and are used here as additional support for the interpretation of Mn/Al and Mn/Ca records. Reactive phosphorus undergoes diagenetic transformation controlled by bottom water oxygenation, and organic carbon flux[38]. Depletion of O₂ in the deep ocean causes the reductive dissolution of P (particularly reactive P bound to ferric iron oxides), hence releasing the sorbed P as soluble reactive P[39] (Supplementary Information). Higher export and primary productivity occur together with high BPWO at Sites U1314 (total organic carbon, biogenic silica and calcium carbonate; Supplementary Information, Supplementary Fig. 5A-C) and 1094 (XRF Ba/Fe and Ca/Fe, Supplementary Fig. 1[40]) during early interglacial stages. During the MPT, colder deep-water masses were present in all oceans, and the temperatures of these water masses remained relatively steady during glacial periods[41]. These observations eliminate temperature-driven oxygen-solubility changes as a factor controlling BPWO. Thus, as opposed to changes in the deep-ocean temperature or the supply of organic matter to sediments and its remineralization, which is more likely driven by local processes, we attribute the changes in BPWO in Sites U1314 and 1094 to changes in deep-ocean circulation operating at basin scale and related to global climate processes.

## A poorly oxygenated deep Atlantic across glacials and terminations

Geochemical data from Site U1314 show pronounced shifts interpreted as changes in circulation and bottom water chemical properties between marine isotopic stages (MIS) 26 (970 kyr), MIS 22 (880 kyr) and MIS 20 (780 kyr) (Fig. 2B, C). Before MIS 26, short-lived events with decreased BPWO occurred only during episodes of high IRD delivery at Site U1314 (Fig. 2F). In contrast, the glacial stages and terminal stadial events since MIS 26 – which are characterized by more intense iceberg discharges– result in fresher surface waters, which enhanced surface buoyancy and limit the potential for open-ocean deep convection, as shown by the marked negative δ[13]C$_b$ excursions at Site U1314 (Fig. 2A). During these intervals, lower bulk Mn/Al and P/Al, planktonic foraminifera Mn/Ca and reactive P phases suggest a more sustained reduction in BPWO and a loss of dissolved P and Mn from the sediments to the water column (Fig. 2B-C). Decreased BPWO at Site U1314 is supported by the benthic foraminifera assemblages. Deep-sea

species *Astronomion novozealandicum*, *Lobatula wuellerstorfi* and *Gyroidina umbonata* are related to well-ventilated, oxygen-rich bottom waters[42,43]. Abundance of these species follow the same trend as the other geochemical proxies, with pronounced minima during cold periods at 1060 kyr (MIS 30), 960 kyr (MIS 26), between 940 and 870 kyr (MIS 24–MIS 22) and 790 kyr (MIS 20) (Fig. 2D). These times are also associated with enhanced sea ice extent in the North Atlantic[44]. In contrast to the geochemical proxies, benthic foraminifera species indicative of well-ventilated deep waters suggests higher BPWO across MIS 23 (Fig. 2D). Overall, positive δ[13]C$_b$ values during MIS 23 is a common feature in all sites across the Atlantic basin[45] (Supplementary Fig. 2). This evidence suggests that NADW production occurred in some intervals, when freshwater input to the high-latitude North Atlantic was at its minimum (according to the lowest IRD/g values between 915–911 kyr and between 900-890 kyr), and the AF was much northern and sea ice cover was absent/minimal (as indicated by decreases of 50-40% in the polar and sea-ice related microfossils *N. pachyderma* and *C. davisiana*, respectively). Therefore, the discrepancy between geochemical and faunal records during MIS 23 at Site U1314 is explained by the different sensitivity of these proxies to BPWO levels[46]. Bulk Mn/Al and P/Al ratios, reactive P and the relative abundance of benthic fauna with high-BPWO affinity also show lower values during the onset of interglacial phases during MIS 25 and MIS 21 (Fig. 2B-D). This suggests that deep-ocean ventilation via NADW production did not resume immediately after ice-volume decreased. Overall, BPWO proxies show significant correlations among them, as well as with other proxies indicative of changes in surface oceanography, which point to a southward advance of the AF with the consequent increase in freshwater and sea ice (Supplementary Fig. 6).

A decreased in BPWO at Site U1314 during the MPT is supported by pronounced negative δ[13]C$_b$ excursions during glacial stages across the MPT; particularly during MIS 24 and MIS 22, at deep sites (> 2500 m) across other high-latitude North Atlantic sites bathed by the NADW today (e.g., 980/1, 983, and 984; Supplementary Fig. 2)[45,47], while shallower sites record increased ventilation at intermediate depths (Site 982)[48]. The concomitant low δ[13]C$_b$ water reflects advection of nutrient-enriched deep water, which could be attributed to upstream hydrographic changes, replacement or mixing with southern source waters or local water-mass reorganizations. Hydrographic changes in the Nordic Seas that lead to nutrient-rich waters being flushed-out downstream during the MPT has been discarded by Kleiven et al.[49] based on δ[13]C$_b$ and sortable silt records at Site 983, which is at a similar location and depth as U1314. A widely accepted hypothesizes to explain the low–δ[13]C$_b$ signal in sediment cores in the high-latitude North Atlantic at depths below 2500 m is the increased contribution of oxygen-poor southern-sourced waters to the Atlantic which mix or even replace a waning NADW[6,45,47]. Moreover, at Site U1314 changes in δ[13]C$_b$ and bulk Mn/Al are coeval with fluctuations in XRF K/Ti (Supplementary Fig. 7A). Low XRF K/Ti values correspond to basaltic material (Ti rich) delivered from the Iceland-Faroe Ridge by a vigorous ISOW flowing southward[50]. In contrast, high XRF K/Ti values occur at the same time as the negative δ[13]C$_b$ and low bulk Mn/Al values during glacials and millennial-scale

cold events, indicating a shoaling of the ISOW and a different sediment source region[50], which would suggest that Site U1314 became influenced by a different deep-water mass.

Alternatively, a shift in the deep-water convection south of the SAF, to the Rockall Plateau and potentially including the Iceland and Irminger basin, has been also hypothesized for the Last Glacial Maximum (LGM)[51–53] and MIS 12[54] to argue transformations in deep-water mass characteristics of northern-sourced waters rather than an increased admixture of southern-sourced water. Under this scenario, the glacial $\delta^{13}C_b$ depletion observed in sediment records between 50–60°N and at > 2500 m depth in the North Atlantic would reflect higher preformed nutrients (due to bottom-water formation under sea ice and/or shelf ice) and increased residence time of northern-source water[49,51], which would lead to lower BPWO. More recent studies support this mechanism by using stable isotopes and εNd to trace deep-water mass changes which suggest uninterrupted (but weak) NADW formation during the LGM[55–57]. Although the current dataset available at Site U1314 does not allow an unambiguous identification of which water mass was responsible of decrease in BPWO, the lower $\delta^{13}C_b$ during glacial intervals characterized by decreased BPWO at Site U1314 would correspond to a deep-water mass that had long been isolated from atmospheric exchange. Therefore, we suggest that the increased storage of respired carbon in the high-latitude North Atlantic was due to a weakening of the AMOC and shoaling in the boundary between AABW and NADW during glacial intervals of the MPT.

The evolution of BPWO proxies and $\delta^{13}C_b$ at Site U1314 is consistent with nutrient, paleoxygenation and deep-circulation proxies from downstream sites along the Atlantic basin. A 60-40% increase in dissolved phosphate estimates from the North [DSDP Site 607; 10] and South [ODP Site 1267; 8] (Supplementary Fig. 7B) and a decreases in the P/Al ratio at IODP Site U1308 (Supplementary Fig. 7C) between MIS 24–22 coincides with lower reactive P and P/Al at Site U1314, which suggests the build-up of a deep ocean nutrient inventory and higher respired carbon pool in the deep Atlantic Ocean. The timing of the decrease in Mn/Al and P/Al ratios and reactive phases displayed by the U1314 records is also coeval with the largest decreases in BPWO at IODP Site U1385 based on the carbon isotope gradients between epi- and infaunal benthic foraminifera ($\Delta\delta^{13}C_{wue-aff}$) (Fig. 2F), located downstream from the convection areas in the Nordic Seas and North Atlantic[23] (Fig. 1C). Modest decreases in $O_2$ (~5–15 μmol/kg) at Site U1385 (SW Iberian Margin) occur simultaneously with small IRD peaks at U1314 during MIS 30 (1053 kyr) and MIS 28 (1012 kyr) (Fig. 2F). During these glacial stages, BPWO proxies at Site U1314 show more pronounced decreases, indicating either different sensitivity of these proxies to oxygen concentrations compared to that observed at U1385, or a different magnitude in oxygen changes at both latitudes (Fig. 2B-D, F). Later, $\Delta\delta^{13}C_{wue-aff}$-based $O_2$ estimates at U1385 and IRD peaks at U1314 show larger amplitude changes since MIS 26 and particularly during MIS 24, MIS 22 and MIS 20. In contrast to Site U1314, the $\Delta\delta^{13}C_{wue-aff}$-based $O_2$ estimates at Site U1385 appears to indicate a more abrupt transition between suboxic to oxic conditions, which could be related to the nature of the proxy ($\Delta\delta^{13}C_{wue-aff}$ is truncated above 235 μmol kg$^{-1}$[23]) or point to different water mass signals. εNd records, a deep-water mass provenance indicator, in the mid-latitude North and South Atlantic, suggest a substantial[19–21] or moderate[11] shift in deep ocean circulation (depending on the record resolution) across the MPT, which would be indicative of a weakened AMOC and/or expansion of southern source waters which facilitated increased carbon storage in the deep ocean. Thus, the combined records suggest a connection between the magnitude of IRD deliveries (and related freshwater delivery to the surface ocean), a reduction in deep-water formation in in Greenland, Norwegian and Iceland convection areas and in BPWO in the high-latitude North Atlantic (Site U1314) and downstream reductions in deep-water oxygenation across the Atlantic basin (e.g., U1385; 607).

## Role of cryosphere-ocean interactions on BPWO in the North Atlantic

The two periods with lowest BPWO as inferred from our proxies occurred during the times of expanded ice volume in the northern hemisphere, during MIS 24 and MIS 22[7,15] (Fig. 3A). Lower sea level during these glacial stages may have also allowed the advance of marine-based ice sheets onto the continental shelves[7,15]. A larger ice mass on the northern continents makes the ice sheets more prone to summer melting, releasing meltwater to the high latitude ocean (more IRD)[29], which disrupts deep water formation and slowed down the AMOC[58]. Freshwater injection into the Irminger basin (in the vicinity of Site U1314) might have exerted a pronounced AMOC change compared to other deep water convection regions in the high-latitude North Atlantic[59]. The weakening of the AMOC leads to a shallowing of the mixed layer depth, decreased production and shoaling of the NADW, leading ultimately to a reduction in the oxygen supply to the deep ocean. Cross-correlation analyses of Site U1314 proxy data (see Methods) show no lag between the IRD supply and the response of BPWO and $\delta^{13}C_b$, reflecting the strong and instant coupling between meltwater perturbations response in the deep-ocean (Supplementary Fig. 8). One exception is the ~1 kyr lag of the benthic foraminifera assemblage related to well-ventilated waters (correlation only marginally higher than without any lag; Supplementary Fig. 6), which is similar to the lag between the response of benthic organisms in the North Atlantic and changes in deep-ocean conditions, and it is explained by a series of biotic interactions (e.g., species competitions) that take ~1 kyr to equilibrate[60]. In addition, a weakened AMOC contributes to a reduced northward heat transport and promotes the southward migration of the SAF and expansion of sea ice (as indicated by the abundance of polar and sea-ice related microfossils[30], (Fig. 2E), which occurs with a lag of ~1kyr relative to the freshwater perturbation (Supplementary Fig. 8). This sea-ice cover, if extending over deep-convection regions, would lead to a reduction in the capacity of the surface ocean to exchange gases with the atmosphere[61], further isolating the deep ocean.

Poor BPWO conditions continued during the onset of interglacial phases at MIS 25 (953–945 kyr) and MIS 21 (859–853 kyr), 5–7 kyr after the interglacial maxima (minima in $\delta^{18}O_b$ values) (Fig. 2B–D). This indicates that persistent sea ice and/or freshwater conditions delayed the reactivation of deep ventilation and thus maintained low BPWO in the subpolar North Atlantic into the interglacial stage. Therefore, the prolonged interval of poor deep ocean oxygenation in the subpolar North Atlantic is related to the expansion of ice sheets and meltwater release near the areas of deep-water formation and associated ocean feedbacks during glacial maxima and termination events during MIS 26, MIS 24 and MIS 22, as well as during early MIS 25 and MIS 21. Conversely, the reduction in meltwater supply to the surface ocean and northward retreat of the SAF and sea ice, allowed persistent northward flow of subtropical warm, salty water which increased sea surface density to the point of resuming NADW production, ventilating the subpolar North Atlantic and increasing BPWO.

## A bi-polar interplay driving deep-ocean carbon storage

The freshening of the upper ocean and decreased BPWO in the subpolar North Atlantic at Site U1314 occurred roughly in step with the change toward a freshening and reduced BPWO in the Southern Ocean. Authigenic Mn/Ca peaks in planktonic foraminifera coatings at ODP Site 1094 during glacials remain low except during terminations XII (MIS 26/25) and X (MIS 22/21), reflecting overall low BPWO conditions south of the APF for the entire MPT (Fig. 3G). Colder and fresher surface ocean waters and expanded Antarctic sea-ice cover[12,18,62] (Fig. 3E, F), possibly enhanced water column stratification in the Southern Ocean between MIS 24–MIS 22. This stratification caused an equatorward shift of the APF and the southern westerlies[40], which

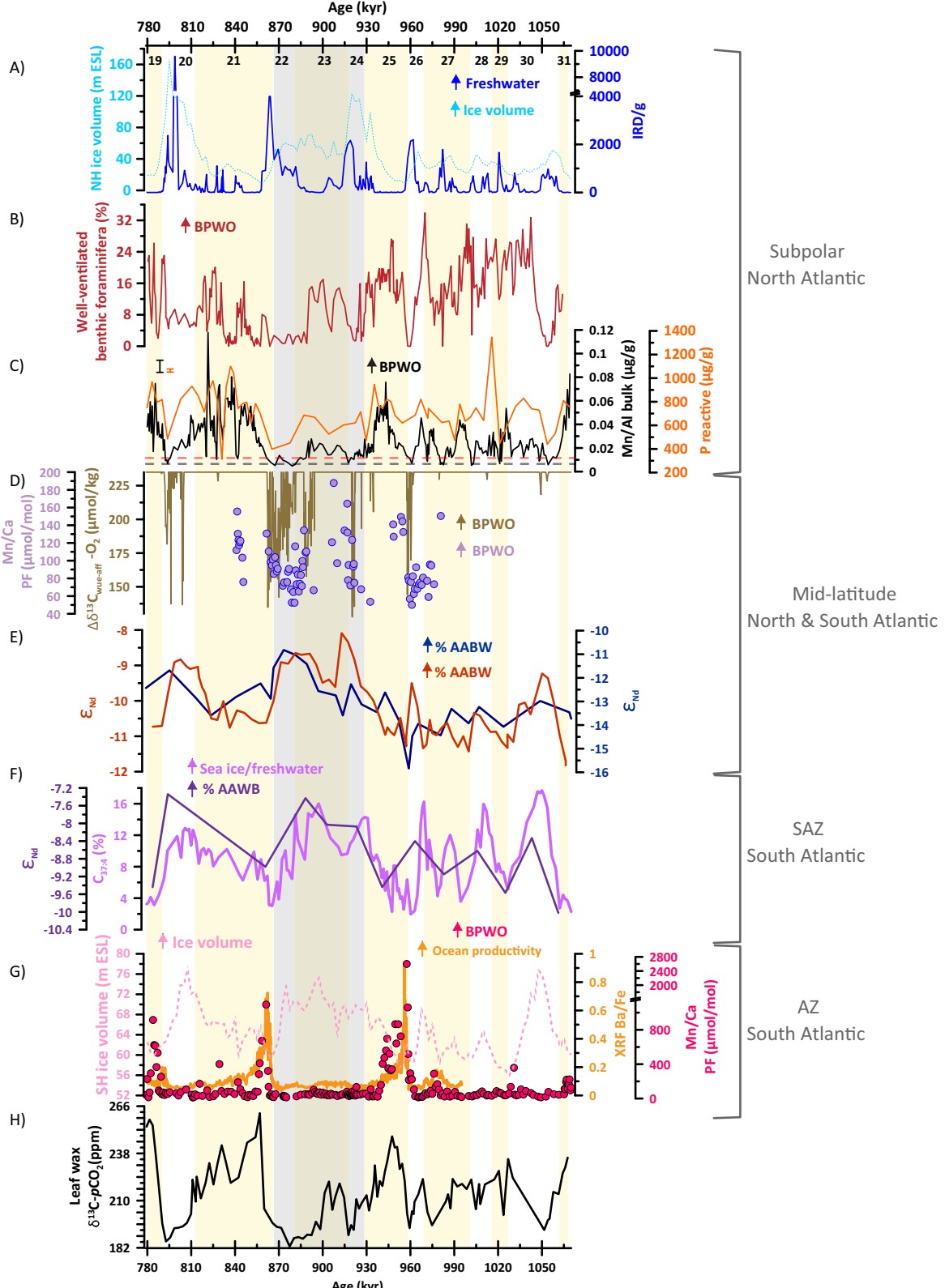

limited ocean ventilation and reduced BPWO as shown by the low planktonic foraminifera Mn/Ca values at Site 1094 (Fig. 3G). In contrast, the peaks in XRF Ba/Fe[40] and higher planktonic foraminifera Mn/Ca values at the same site during terminations XII and X indicate Antarctic warming, poleward retreat of the APF and sea-ice cover and the return to oxic conditions and $CO_2$ outgassing in the Southern Ocean via resumption of deep ventilation in this region.

The enhanced freshwater input to the surface ocean in both polar regions resulted in enhanced stratification that prevented efficient deep ocean convection by changing deep ocean density. However, a cooler surface ocean with extensive sea-ice formation in the Southern Ocean could have caused a more efficient brine export to the deep ocean[63], enhancing the density of the AABW formed in the Southern Ocean[64]. The weakening and shallowing of the AMOC upper cell in the

**Fig. 3 | Changes in ocean feedbacks and climate in both Atlantic polar regions across the MPT.** North Atlantic IODP Site U1314 surface and deep-water records (this study) compared with AMOC, sea-level and $pCO_2$ estimates across the MPT. Black numbers indicate interglacials (odd) and glacials (even) marine isotope stages (MIS). Yellow vertical bars correspond to interglacial stages. **A** IODP Site U1314 IRD/g (dark blue line)[28] and Northern Hemisphere ice-volume reconstruction (dashed light blue line)[15], (**B**) deep-sea benthic foraminifera species related to well-ventilated bottom waters (dark red line), (**C**) bulk Mn/Al ratio (black line) and P reactive (orange line). Error bars are replicate 1 SD (Methods). **D** Site U1385 $O_2$ concentration based on $\Delta\delta^{13}C_{wue-aff}$ and Mn/Ca ratio (golden line) in benthic foraminifera (*Uvigerina* spp.) authigenic coatings (purple circles)[23]. The $O_2$ reconstruction is truncated at 235 μmol kg⁻¹, above which the calibration is unreliable. **E** Site 607[21,22] (dark blue line) and Site U1479 εNd[11] (brown line). **F** Site 1090 εNd[19] (dark purple line) and relative abundance of $C_{37:4}$[88] (light purple line). **G** Site 1094 sedimentary Ba/Al measured by core scanning XRF[40] (light orange line) and Mn/Ca ratio in planktonic foraminifera (*N. pachyderma*) authigenic coatings (pink dots), where Mn/Ca > 100–200 μmol/mol can be indicative of potential Mn-Fe coating material [17; note that axis is truncated between 1200 and 1700] and Southern Hemisphere ice-volume reconstruction (dashed pink line)[15]. **H** Site U1446 reconstructed atmospheric $CO_2$ concentration ($CO_2^{FA}$) based on fatty acid $\delta^{13}C$ (black line)[5]. The vertical gray bar highlights MIS 24–MIS 22.

subpolar North Atlantic and vertical density gradient compared to the AMOC lower cell, facilitated that the AABW filled the deep ocean[65]. The glacial expansion of the AABW provided a more extensive deep ocean reservoir for carbon storage[11,61] which would help to explain the moderate increase in $pCO_2$ during MIS 23 compared to MIS 25 and MIS 21 and the low glacial $pCO_2$ values particularly during MIS 22 and MIS 24 (Fig. 3H). The presence of more nutrient-enriched/oxygen poor glacial deep water (> 2500 m) is inferred from increasing trends in deep ocean nutrient and dissolved inorganic carbon concentrations[8,26,45,47] across the entire Atlantic ocean between MIS 24–MIS 22 (Supplementary Fig. 2). These findings are also consistent with a recent study using new high-resolution paired εNd (Fig. 3E-F), carbon and oxygen isotope data from the South Atlantic Ocean[11], which suggest a shoaling in the boundary between AABW and NADW as the main mechanism increasing storage of respired carbon during glacial intervals of the MPT. Our results showing decreased BPWO in the high-latitude North Atlantic during glacial intervals of the MPT (particularly between MIS 24 and MIS 22) support previous analyses that also suggested a shallow glacial AMOC, driven by a reduction in the relative amount of NADW formation in the high-latitude North Atlantic and/or increasing the relative volume of southern source water (Fig. 4)[8,11,19,20]. At the same time these results illustrate the need of paleoxygenation studies with additional proxies such as εNd or $[CO_3^{2-}]$ reconstructions for accurate deep-water mass tracing and mixing estimates[55]. One important observation from the records of both polar regions is that the trends in BPWO during glacial terminations differ in timing between the two poles, particularly during Terminations XII, X and IX. The recovery to high bulk sediment Mn/Al and P/Al ratios and high BPWO-fauna at Site U1314 is more gradual than at Site 1094 and maxima in BPWO appear only after attaining full interglacial conditions (5–7 kyr lag relative to the Ba/Fe peaks at Site 1094; Fig. 3C, G). The different timing of oxygenation recovery between sites U1314 and 1094 should not be a chronological artefact, since benthic isotope records from both sites are aligned to the LR04 benthic stack (Supplementary Fig. 2). This lag is too large to be solely attributed to the residence time of the water masses in the deep Atlantic and Southern Oceans [only few hundred years at maximum[66]. Instead, we propose that this lag could be attributed to a larger magnitude of initial freshening in the convection areas of the Northern Atlantic (based on the larger IRD peaks since MIS 26) that delayed the initiation of the AMOC[67], and to the different regional thresholds driving phasing of IRD peaks[62] (and thus freshwater supply) and deglacial warming (and thus BPWO response) in the Southern Ocean and North Atlantic[68].

Our study suggests that the Atlantic ocean's interior in both polar regions underwent analogous changes in BPWO in both polar regions during the MPT. While proxy records from the South Atlantic and Southern Ocean provide a plausible explanation of climate feedbacks which could be potential precursors to, or even a direct trigger of the MPT[8,11,12,16,17,40], we provide critical evidence that demonstrates the unrecognized importance of the physical processes (deep-water formation, freshwater release, sea-ice formation) in the subpolar North Atlantic in triggering abrupt AMOC oscillations and shoaling the boundary between north and southern source waters, which are fundamental in the modification of the oceanic carbon storage during the MPT. The increased carbon trapping in the deep ocean would modify the carbon cycle, thereby contributing to the transition from the 41-kyr world, characterized by a linear response to insolation forcing, to the nonlinear 100-kyr world[4,69]. Climate model experiments predict a marked oxygen reduction in the deep ocean for the North Atlantic in the near future under a potential future AMOC reduction scenario due to climate change[70]. Considering these predictions and our results, we advocate for expanded paleoxygenation studies focused in the subpolar North Atlantic during warmer periods of Earth's history, as it may provide critical of the understanding the dynamic mechanisms driving global carbon storage and ocean deoxygenation over longer timescales.

## Methods

Samples used in this study were recovered during International Ocean Drilling Program Expedition 306. Site U1314 was cored in the Gardar Drift on the eastern flank of the Reykjanes Ridge, in the subpolar North Atlantic. The age model and benthic foraminifera stable isotope data was previously published in Hernández-Almeida et al.[29] and it is made by aligning the benthic foraminifera $\delta^{18}O$ record, using mainly *Lobatula wuellerstorfi* (formerly "*Cibicidoides*" *wuellerstorfi*", https://www.marinespecies.org/aphia.php?p=taxdetails&id=1746350) and *Melonis pompilioides*, to the LR04 benthic stack[71].

The Mn/Ca ratios in planktonic foraminifera for sites U1314 and 1094 were previously published, prepared and measured via similar cleaning protocols, which include clay removal and oxidative and reductive steps[17,72].

### Elemental geochemistry

A total of 298 samples at approximately 8 cm resolution spanning 60–84.16 mcd were analyzed using total sediment digestions of bulk sediment for major/minor elements. Approximately 0.1 g of powdered sample was dissolved by adding 10 ml of purified Mili-Q water and concentrated trace-metal grade $HNO_3$, HF and HCl, following EPA SW846 Method 3050. Total digestion was performed in a CEM MDS-2000 microwave. Once the digestion was complete, boric acid was added to stabilize the solutions. The samples were transferred to new polypropylene tubes and diluted to 50 ml of Milli-Q water. Total elemental concentrations were determined using a P950 Inductively Coupled Plasma - Atomic Emission Spectrometer (ICP-AES) with a CETAC AT5000+ Ultrasonic Nebulizer in the Department of Earth and Environmental Sciences at Indiana University. Elements analyzed for total concentrations include P, Al, Mn, Ca and Fe. National Institute of Standards & Technology (NIST) Standard Reference Material (SRM) 1646a, estuarine sediment, was used to evaluate analytical reliability. The certified value for P is 0.027 ± 0.001.

We normalize the concentration of Mn and P in bulk sediments relative to Al, based on the conservation behavior of Al in marine sediments[73]. Mn/Al ratios (0.004–0.117) are generally in excess during interglacial intervals of what can be expected from the average crustal

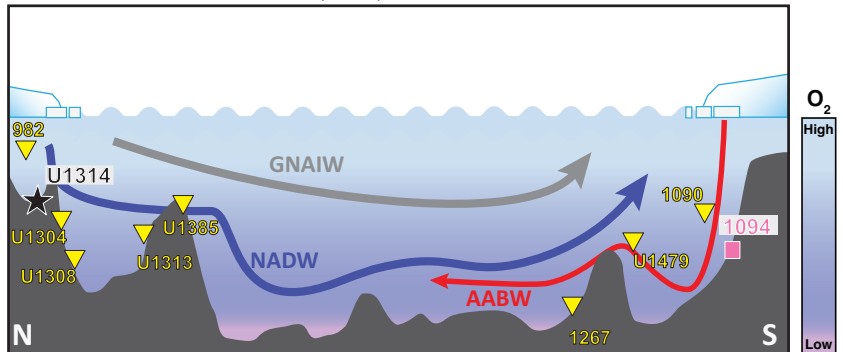

A) Glacials and IRD events before MIS 26 (960 ka).

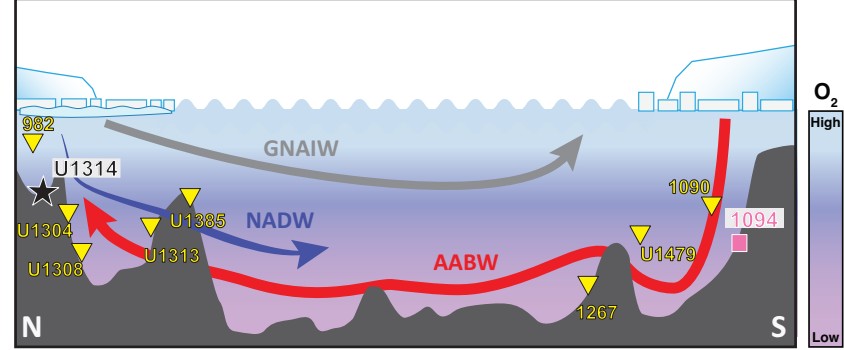

B) Glacials and IRD events after MIS 26 (960 ka). Northward intrusion with southern source waters

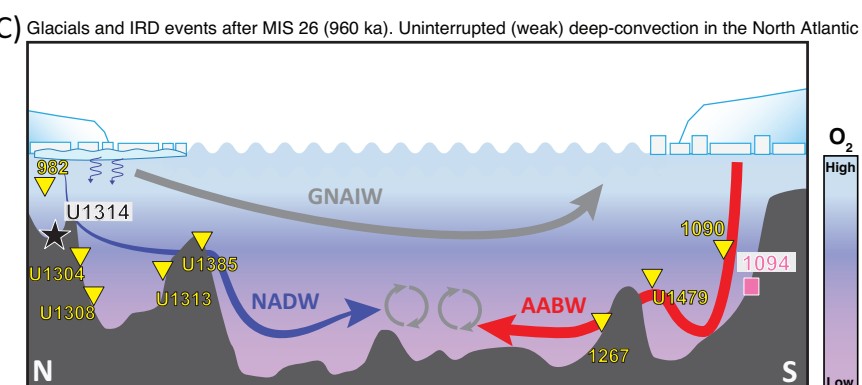

C) Glacials and IRD events after MIS 26 (960 ka). Uninterrupted (weak) deep-convection in the North Atlantic

**Fig. 4 | Schematic illustrations of the changes in the Atlantic basin across the MPT during. A** Glacials and IRD events before MIS 26 (960 ka), and (**B**), (**C**) two possible scenarios for glacials and IRD events after MIS 26 (960 ka). The glacials before MIS 26 (1069–960 ka) are characterized by a lower NADW production in the high-latitude North Atlantic, but without dramatics reductions in BPWO (as observed in BPWO records from Sites U1314 and U1385). One exception is MIS 30, which is characterized by a moderate reductions in bottom-ocean $O_2$[23]. The post-MIS 26 period is characterized by a much larger ice-volume in both polar regions[7,15], sea-ice and surface ocean cooling[28,88], and an unprecedent reduction in BPWO as shown by Site U1314 and Site U1385 proxies records which reflects the weakening of the AMOC[19]. The main feature is the shoaling in the boundary between AABW and

NADW, and the strengthening in mid-depth circulation via formation of glacial North Atlantic intermediate waters (GNAIW)[48]. Two potential scenarios are proposed: (**B**) Northward intrusion of southern source waters, as suggested by benthic carbon isotope records[6,45,47] from the Atlantic basin for the MPT; (**C**) Uninterrupted but weak deep-convection in the North Atlantic, with NADW formed in isolation from the atmosphere (under sea-ice) and/or with longer residence time, following recent models of thermohaline circulation in the high-latitude North Atlantic proposed for the LGM[55,56]. Small blue arrows in (**C**) denote deep-water convection below sea-ice. Large Arrows indicate the major deep and intermediate ocean water mass flows, with thin arrows associated with NADW and AABW representing an inferred lower flux of these water masses during glacials.

abundance (i.e., 0.0075 for Upper Continental Crust (UCC); 0.0116 for Average Andesitic Crust (AAC))[74]. During glacial intervals (e.g., 888–866 kyr) or short-lived cold intervals (at 1060, 1000 and 980 kyr), the Mn/Al ratios are close to the average detrital composition (Mn/Al <0.011). External precision was evaluated using 20 randomly chosen samples, with 1 SD replicate precision of ±120 µg/g for Mn/Al, 0.0098 µg/g for P/Al.

## Total phosphorus extraction

Sequential extractions to determine sedimentary P components were performed on a subset of 64 samples at 32–42 cm intervals using the procedure outlined by Filippelli and Delaney[75] after Ruttenberg[76], with the modification that adsorbed and iron-bound P were combined into one step, dithionite-extractable or oxide-associated P (Table 1). The sequential extraction technique yields chemical information about the

**Table 1 | Extraction steps of the sequential extraction of P [after[75,76]]**

| Step | Reagents | P component isolated |
|---|---|---|
| Oxide-associated | 10 mL CDB solution (6 h) (0.22 M Na citrate, 1 M NaHCO$_3$, 0.13 M Na-dithionite), 10 mL of 1 M MgCl$_2$ (2 h), 10 mL H20 (2 h) | Adsorbed and reducible or reactive Fe-bound P |
| Authigenic | 10 mL of 1 M Na-acetate buffered to pH 4 w/acetic acid (4 h), 10 mL of 1 M MgCl$_2$ (2 h)-twice, 10 mL H$_2$O (2 h) | Carbonate fluorapatite (CFA), biogenic hydroxy apatite, and CaCO$_3$-bound P |
| Detrital | 10 mL of 1 M HCl (16 h) | Detrital P |
| Organic | 1 mL 50% (w/v) Mg(NO$_3$)$_2$, dry in low oven, ash at 550 °C (2 h), add 10 mL of 1 M HCl and shake (24 h) | Organic P |

reactivity of phosphorus within sediments; the geochemical fractionation leads to the apportionment of phosphorus in several operationally-defined pools[76]. Reactive phosphorus pools, so-called because the phosphorus has the geochemical potential for release to the environment, include organic phosphorus, adsorbed and oxide-bound phosphorus, and authigenic–biogenic phosphorus.

Approximately 0.1 g of dried and ground sample was weighed into new 15 mL polyethylene centrifuge tubes. Samples with reagents were shaken on an orbital shaker for the recommended amount of time depending on the protocol for each extraction step and then centrifuged for 10 min. All supernatants were decanted into acid cleaned polyethylene bottles, pooled and saved for analysis. A Shimadzu scanning UV-Visible Spectrophotometer at Indiana University was used for the determination of P concentrations for authigenic P, organic P and detrital P (steps II–IV) from the sequential P extraction using the molybdate blue technique for color development[77]. Dithionite-extractable P (step I) concentrations were determined by ICP-AES. Each sample were measured at least twice for each of the P phases, and the reproducibility of all phases was on average ±13 μg/g (1 SD).

**Total carbon**

The total carbon (TC) content of the sediment was measured in 584 samples using a UIC Coulometrics CM150 carbon analyzer at Indiana University. For total organic carbon (TOC) analyses, first we removed the total inorganic carbon (TIC) following standard procedures[78]. About 0.2 g of powdered sample was acidified with 2 N HCl in 50-mL. Samples were dried overnight, rinsed with deionized water, centrifuged and decanted two times. Once dried, TOC was then measured using a Flash 2000 Combustion CHNS/O analyzer.

**Benthic foraminifera**

Sediment samples for benthic foraminifera analysis were collected approximately every 8 cm between 60- and 83.76-mcd, which correspond to the time interval between 780 and 1064 kyr. The average sample resolution is ~1 kyr. Samples were dried, weighed, wet sieved over a 63 μm mesh sieved and the residue weighed again after drying. Benthic foraminifera were picked, identified and counted from dry residues in the >125 μm fraction at Universidade de Vigo. Each sample was split to obtain a minimum of about 200 individuals. Foraminiferal identifications follow the taxonomic concepts of WoRMS (last accessed on 2024-05-28)[79] Images of the species described in this manuscript are shown on Supplementary Fig. 9.

**Statistical analyses**

Records were linearly interpolated at 0.97 kyr (similar to the average sample resolution of benthic foraminifera census counts and inorganic geochemistry) using *approx* and *Map* functions as implemented in "stats" and "purrr"[80] packages, respectively. The linear correlation was performed using the functions *chart.Correlation* and the cross-correlation using function *ccf*, from the packages "PerformanceAnalytics"[81] and "tseries"[82], respectively. All analyses were made using R[83]. We calculated temporal offsets between meltwater input (IRD/g) and surface ocean temperature and sea-ice conditions (relative abundance of *Neogloboquadrina pachyderma* and *Cycladophora davisiana*), deep circulation (δ$^{13}$C$_b$) and BPWO (bulk Mn/Al and P/Al, benthic foraminifera species related to well-ventilated waters) changes between 1069–779 kyr. Reactive P was excluded of this analysis because its original sample resolution was four times lower. We then computed the cross-correlation between records at a given lag (offset at which highest correlation occurs). Negative (positive) values indicate that IRD delivery occurs earlier (later) than a given proxy.

## Data availability

The datasets generated during the current study are available as Supplementary Data 1–4 and have been archived in PANGAEA (https://doi.pangaea.de/10.1594/PANGAEA.980512; https://doi.pangaea.de/10.1594/PANGAEA.980516; https://doi.pangaea.de/10.1594/PANGAEA.980506; https://doi.pangaea.de/10.1594/PANGAEA.980480).

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

## Acknowledgements
We thank the International Ocean Discovery Program (IODP) for providing samples Site U1314 core samples. Thanks to Leopoldo Pena for commenting an early version of the manuscript. We also thank Angela K. Robertson and Rosalice Buehrer for their help in the lab.

## Author contributions
I.H.A. and P.D. conceived the study. P.D. performed benthic foraminifera census counts. I.H.A. performed the geochemical analyses with the support of GMF. I.H.A. and P.D. analyzed the proxy data and wrote the manuscript with contributions from F.J.S., G.M.F., and A.H.L.V.

## Funding

## Competing interests
The authors declare no competing interests.
