## [Transparent Peer Review file · Nature Communications]

Glacial dysoxia in the deep subpolar North Atlantic during the Mid-Pleistocene Transition

Corresponding Author: Dr Iván Hernández-Almeida

Version 0:

Reviewer comments:

Reviewer #1

(Remarks to the Author)

Hernandez-Almeida et al. have pulled together an interesting set of data to try to understand bottom water and pore water oxygenation and how that relates to deep water circulation. The data is interesting and of high quality. Ultimately, I do think the data and interpretations merit publication, but I think the manuscript requires some significant modifications. In particular, I think that the figures could be constructed in a way that is less confusing. For example, some of the data is presented in multiple figures while the reader is forced to compare at times between figures. For example, line 152 references Figs. 1, 2 and supplementary Fig. 4, but each of these figures has multiple parts, and I have no idea which parts I am supposed to look at. In addition, there are a lot of broad statements about the MIS 24-22 transition, for example there are multiple places where the the interval from 940-870 kyr is discussed with respect to several proxies. The authors suggest that the proxies demonstrate that there is low BPWO across this interval, but the benthic foram ventilation proxy does not agree with this statement. Fig. 2D/3B suggests relatively higher BPWO across MIS 23 in contrast to the other proxies that suggest low BPWO across the entire interval from MIS 24-22. The discrepancy between these proxies needs to be explained. All of the geochemical proxies do agree. The difference between the benthic forams and the geochemistry needs more attention.

Other comments:

1. Some of the sentences are too complicated. Sentences should include one idea. There are also some awkward phrases, such as "converge in the idea" line 50.
2. I wouldn't call an authigenic mineral a contaminant - line 102.
3. Some of the colors used in the figures are difficult to differentiate (Fig. 2E and 2F, Fig. 3G)
4. Figure 1 is not easy to read.
5. Table 1, there is no time indicated for the Na-acetate solution or for the organic step.
6. The way the figures were constructed is a little bit confusing and the reader is asked to move back and forth between fig. 2 and 3 and figures in the supplemental information.
7. The reader is referred to the MIS for Fig. 3, but these are not listed on the figure.
8. Figure 2E caption needs to include a mention of these taxa being sea-ice related as indicated in the text line 175. Also, the trends in the figure and relationship to other data is not clear.
9. From the graphs, I don't see evidence of low BPWO in MIS 21 or 25.
10. It is very confusing to refer to MIS when that information is not available for all figures.
11. There are numerous places where ", and" occurs when the "," is not needed.
12. I think the method is actually Method 3050, not 305 (line 267).
13. I gram of sediment in a 15 mL centrifuge tube seems like a lot (line 299). 0.1 g was used for the total digestion, should 0.1 g also be used for the sequential P extraction?
14. Instead of saying Mn spikes, perhaps maxima? (Line 15 of supplemental information and elsewhere)
15. By phosphorus provenance (line 70 supplemental information) do you mean the different pools of P? This is a confusing use of provenance. Presumably reactive P was once related to organic matter.
16. Sorry these last comments are out of order. Line 41 states that the deep ocean is the largest C reservoir, but it actually sedimentary rocks.
17. Line 77 - it would be helpful to identify the two sites downstream by site number.

(Remarks to the Author)

This manuscript provides an insightful analysis of bottom water oxygenation (BPWO) changes across the subpolar North Atlantic during the mid-Pleistocene transition (MPT), emphasizing the role of ice sheet dynamics, freshwater input, and Atlantic Meridional Overturning Circulation (AMOC). The authors reconstruct BPWO using multiple proxies—including solid-phase manganese (Mn) enrichments in bulk sediments and planktonic foraminifera coatings, sedimentary phosphorus (P) reservoirs, and benthic foraminiferal assemblages—at IODP Site U1314. By integrating previously published records, the study highlights the role of cryosphere-ocean coupling in the North Atlantic and Southern Ocean in controlling ventilation, BPWO, and carbon storage, potentially offering new insights into the mechanisms of the MPT.

However, several issues should be addressed to improve the manuscript. While the use of multiple proxies is a strength, the technical details require further clarification, and the discussion could better incorporate the limitations of each proxy. Additionally, the structure and flow of certain sections, particularly the methods and supplementary information, could be improved for clarity. Most importantly, the discussion of the mechanisms needs further clarification.

Major comments/suggestions are as follows:

1. Introduction:

While the introduction effectively synthesizes previous studies, it should better highlight the originality and purpose of this study. The research gap addressed by the authors should be explicitly stated.

2. Methodology:

The use of multiple proxies is a strength of this study; however, the methodological details require further clarification. Clearly distinguish which proxies were measured by the authors and which were obtained from previous studies. Please provide a more detailed description of proxy and potential diagenetic effects to improve transparency. For example, clarify how Mn/Al and P/Al were validated against independent redox indicators, and the potential post-depositional alterations of P phases and their impact on interpretations. Both the methods and Supplementary information need restructuring.

3. Use of Mn Proxies for BPWO: The text describes how Mn/Al and Mn/Ca serve as proxies for bottom and porewater oxygenation. However, it would be useful to clarify how these proxies compare with other commonly used redox-sensitive elements (e.g., U, Mo, V). Are there any limitations or uncertainties associated with Mn-based proxies in these records?

4. Proxy interpretations: Most proxies indicate lower BPWO during the MPT; however, discrepancies among them need further explanation. For example: while Mn/Al suggests generally lower BPWO during the MPT (Fig. 2B), well-ventilated forams only show lower abundance during MIS 24 and 22 but higher abundance during MIS 23 (Fig. 2D). The reasons behind these inconsistencies should be explicitly addressed. Consider alternative interpretations and discuss the robustness of each proxy.

5. Mechanisms: The authors argue that BPWO in the North Atlantic and Southern Ocean was primarily controlled by the ventilation, but other potential influencing factors (e.g., temperature-oxygen solubility and productivity-driven oxygen consumption) should be addressed. If ventilation was the primary control on BPWO, clarify why North Atlantic ventilation was specifically driven by freshwater forcing from iceberg melting. Further elaborate on how these processes relate to the broader climatic transition during the MPT. Expand the discussion on the role of Southern Ocean processes and reference additional studies where relevant.

6. Conclusion: The conclusion effectively summarizes the findings but should more explicitly discuss the broader implications for MPT climate dynamics. Additionally, consider explicitly stating whether the findings align with or challenge previous models of deep ocean carbon sequestration during the MPT.

Consider discussing BPWO changes before and after the MPT to provide additional insights into the mechanisms driving the transition.

7. The section on BPWO during the MPT provides a well-structured discussion using proxy records from IODP Site U1314 and ODP Site 1094. The integration of different geochemical proxies (Mn/Al, Mn/Ca, and phosphorus phases) is commendable, as it enhances the robustness of the interpretations regarding bottom water oxygenation. Consider briefly discussing why these sites were chosen specifically to study BPWO during the MPT, especially in the context of past studies. More explicitly compare BPWO trends between sites U1314 and U1385 is needed, particularly in relation to $\delta^{13}\text{C}_b$ signals.

8. Figures should be explicitly referenced in the text. When citing figures, specify which curve is being discussed (e.g., Fig. 3A rather than simply Fig. 3). Additionally, consider adding a figure synthesizing the hypothesized sequence of events connecting freshwater forcing, ventilation changes, and carbon sequestration. Also, supplementary references should be more clearly tied to the discussion. Instead of simply citing them, briefly indicate what they illustrate and how they support the argument.

More specific comments:

Line 60: Specify which hypothesis is being referenced?

Line 77: Clarify which two sites are being referred to.

Line 80: Provide a reference?

Line 86-89: Expand on how the interpretation of lower $\delta^{13}\text{C}_b$ at Site U1314 distinguishes between reduced NADW ventilation and intrusion of southern-sourced waters. Are there additional proxies (e.g., Nd isotopes, benthic foraminiferal assemblages) that support one interpretation over the other? Explain the higher $\delta^{13}\text{C}_b$ values during MIS 23.

Line 119-122: Specify which productivity proxies were analyzed and explain why productivity changes did not drive oxygenation changes.

Line 155-158: Justify why BPWO changes are attributable to ice sheet dynamics rather than other factors.

Lines 143-157: The section discussing "A less active deep-water convection..." should better distinguish between local and basin-wide processes.

Line 274: The dataset includes bulk elements such as P, Ca, Al, Ba, Ti, Fe, Mg, but only shows P, Mn, and Al. Consider providing all measured elemental data. Did the authors measure U concentrations? As a redox-sensitive element, U could provide additional insights into oxygenation changes.

Line 326: Address how samples with no well-ventilated benthic foraminifera were handled. Was this based on a minimum count of 200 individuals? Could absence be due to random sampling bias?

Figures 1 and 2 are referenced, but their role in supporting key arguments could be more explicitly stated in the text. For example, how does Figure 2A specifically demonstrate reduced ventilation or changes in nutrient concentrations?

Supplementary information

References to figures and supplementary information should be more explicitly tied to the discussion. Instead of simply citing them, the authors could briefly indicate what these figures illustrate and how they support the argument.

Line 10: "Under oxygen-rich conditions, Mn is then removed from porewater..."—the wording suggests an absolute removal process. Consider specifying that Mn precipitation as Mn-rich carbonates or oxides is a dominant but not exclusive removal mechanism.

Line 34-37: The observed 5–7 kyr offset in Mn peak timing between Site U1314 and Site 1094 is attributed to burial conditions or age model uncertainties. Could other mechanisms, such as regional differences in deep water ventilation, contribute to this offset? This point should be explicitly considered.

Line 36-39: The timing of CO₂ release relative to the onset of well-oxygenated bottom-water conditions requires clarification. Are there any leads or lags between these events? Explicitly addressing this question would strengthen the discussion.

Line 44-46: The conclusion that the Mn peak offset represents a real delay in deep ocean ventilation re-initiation is reasonable but not fully supported. The discussion would benefit from a more cautious wording or additional evidence.

Line 51-55: The difference in Mn/Ca peaks between Sites 1094 and 1090 is linked to burial conditions and initial Mn incorporation into foraminiferal shells. Consider clarifying whether these factors affect only peak amplitude or also timing.

Line 53-55: How much initial Mn precipitated in the foram shell is needed to create this difference? Please quantify if possible.

Line 56-60: The discussion of Mn contamination would benefit from specifying whether Mn/Ca > 100–200 μmol/mol universally indicates Mn-Fe coatings, or if threshold values vary by site and cleaning protocol.

Line 72-74: It would be helpful to specify whether these lower-productivity sites also differ in terms of redox conditions, sedimentation rates, or detrital input, as these factors can influence phosphorus burial and remobilization.

Line 85-87: The authors should elaborate on why this mechanism does not apply to Site U1314. Do sedimentation rates play a role? Is there an alternative explanation?

Line 86: 'episodes of higher productivity', specify the time intervals that correspond to these episodes for clarity

Line 89-105: Instead of using general terms like "redox proxies" or "ocean productivity proxies," the authors should specify which proxies were employed in this study. Additionally, clarify whether the analysis applies to all glacial periods between 1100–780 kyr or specifically to the MPT.

Line 95-99: The comparison between Site 1314 and other locations is useful, but the discussion would benefit from a clearer synthesis of regional differences. Are there factors unique to Site 1314 that influence its productivity-redox relationship differently from these other sites?

Line 109-110: During the MPT, δ¹³C records show higher values during MIS 23. How do the authors interpret this? Does it correspond to stronger ventilation?

Line 110-112: δ¹³C_b is presented as a deep ocean circulation proxy but is then described as sensitive to multiple factors. Consider rephrasing to emphasize that δ¹³C_b is a complex but valuable proxy rather than implying it is unreliable.

Line 113-116: The significant correlation between δ¹³C_b and geochemical proxies is an important point. If possible, briefly state which proxies show the strongest correlation.

Line 118-135: The distinction between *Astrononion novozealandicum* and *A. echolsi* is well-explained, but the connection to the broader study is unclear. Does taxonomic misidentification affect any interpretations in this work?

Reviewer #3

(Remarks to the Author)

Hernandez-Almeida and co-authors investigated the time interval between 780-1056ka - end of MIS31 into MIS19 covering parts of the so-called Mid-Pleistocene Transition - using site U1313 from the subpolar N. Atlantic. They argue that the time interval covering glacial MIS 22 and 24 was a period of special interest because it exhibits very low bottom water d¹³C in benthic forams. These low values apparently reflect dysoxic conditions caused by glacier induced meltwater which then had further implications for the development of the MPT. The rather low epibenthic d¹³C during the interval in question has been noted before (e.g., Thomas et al 2022), but without further investigating the causes. Thus, the authors provide a multiproxy study to tackle the problem which in my view does deserve appreciation. However, there are a number of major problems with their argumentations. There is a need to overcome these to make this study publishable.

The study hinges very much on the rather "traditional" assumption that the glacial-interglacial variability of d¹³C in epifaunal cibs in the North Atlantic reflects the degree of vertical ventilation (bottom water oxygenation or whatever one likes to call it), i.e., the intensity of the AMOC which is dependent on the production of deep water in polar areas such as the Nordic-Labrador seas. But studies showed a strong habitat-related dependency of d¹³C-cibs to organic carbon availability in the ambient water, so-called Mackenson-effect. More recently it became clear that even during times of massive meltwater at the ocean surface (eg, HS1) or times during glacial maxima (eg, LGM) AMOC did not stop but rather continued to function (e.g., Repschläger et al. 2021, Howe et al. 2016). Moreover, in a glacial environmental setting of the subpolar to polar northern regions both planktonic and benthic foram d¹³C signatures are usually strongly depleted during times of enhanced meltwater releases. This would imply that (a) these regions have likely contributed to the d¹³C signature found downstream.

And (b) that southern AABW with its depleted d13C signature – often invoked as evidence for a northward migration during glacials – seems significant only at very large water depths. As we can see in Fig. 1, the investigated cores 1314 (and others, eg, 1313) still belong to the shallower water depth realm.

There is something unusual in the authors arguments about dysoxic bottom-pore water conditions. Why is the d13C record of cibs continuous as one would expect them not to be present at all considering their habitat behavior. Quantitative abundances of infaunal vs cibs would be very helpful in this matter.

Also, like to see a planktonic 18O/13C record as my own experience shows a parallel trend in benthic and planktonic d13C, essentially meaning a clear vertical top-to-bottom (water column) connection. Thus that connection might be well through the food chain with changing surface productivity...etc . And indeed I checked the core's record and found that the authors have actually published (in 2015) that parallel trend into MIS24...! Thus I would like to see a complete 13C record of N. pachyderma and wonder why don't other parts with very low benthic d13C, such as MIS 30, also show the same response...? Of course, to obtain an insight into the pore water and the bottom water, a comparison of d13C between infaunal species with epifaunal cibs seems necessary, perhaps supported by some d13C analyses of the bulk carbon in the sediment (see below).

I also wonder what is the impact of certain type of IRD (ie, carbon-rich sediment clasts) would have on the assumption of stagnated bottom waters? Their bulk TOC record might give insight, because the conclusion of the redox conditions could simply be, by and large, a post-sedimentary pore water process that happened much later and which had nothing to do with conditions at the sediment-water interface at the time of sediment deposition. Between 780-960ka, it is intriguing to see (suppl. Fig 2) calcite and TOC(%) – latter being pretty low in general - to show such similar trends, and in comparison, the Mn/Al bulk mimics the calcite. Why is that?

The paper is based on a lot of “maybes” and other guesses without any further in-depth groundtruthing. This should involve other interval(s) that had a similar glacial environment with very low d13C signature and an indicative benthic foraminiferal assemblage, but for which no dysoxic conditions are implied. For instance, in nearby core 1313 from some greater water depth – this one btw should exhibit the same features during MIS24-22 – MIS 12 would be a suitable test as it also shows very low d13C values (published by co-author Voelker). I do appreciate suppl. Fig 1 but it only provides a crude glimpse into the issue for MIS12 or some other, longer intervals. But besides MIS12 there are many more suitable intervals, especially those from major glacial terminations (see records by Hodell et al. 2025) when the meltwater stratification hypothesis could be further tested.

Having said the above, I find the jump to the southern hemisphere premature and overly ambitious at this stage; the various proxy records selected is a bundle of very different type of “fruit” that might or might not have anything to do with each other or the northern record.

At present I see the investigation and thoughts provided in the manuscript as a solid case study to start off with. However, it would require more in-depth and broader testing of the proxies applied before any globally relevant implications on the ocean-climate system, and here especially the potential involvement of pCO2 during MIS24-22 and the further MPT development, are warranted. The latter one in particular I find really hard to accept at present.

Version 1:

Reviewer comments:

Reviewer #1

(Remarks to the Author)

This is the second time I have seen this manuscript. Overall, I think that it is much improved but could still use some editing. The authors present an interesting interpretation of their data in the context of other published results that does help to explain transitions during the MPT while also pointing to areas where new data is needed. I am satisfied with how the authors have dealt with reviewer comments. Specific suggestions for improvement are listed here:

Beginning line 40 - Last sentence of the first paragraph needs to be edited. Suggestion: The deep ocean, Earth's largest carbon reservoir that readily exchanges with the atmosphere, has been proposed to play a key role in carbon sequestration and could mitigate the climate change risks of global warming.

Line 44 – suggest reorganization: The leading hypothesis to explain the MPT suggests that the initiation of 100-kyr glacial cycles was driven by increased carbon storage in the deep ocean.

Line 45 – The Southern Ocean has emerged as the leading region to explain the climate feedbacks during the MPT.

Line 49 – remove “the” in befor Antarctic

Line 50 – remove “ on the idea”

Line 52 – change to “Enhanced glacial marine carbon sequestration...”

Line 53 – change “deoxygenation” to “reduced oxygen”

Sentence beginning line 56 – add “which” before imply or some other rewording is necessary

Lie 62 – “should” or “would”? then “would” becomes “could”?

Line 82 – “New and previously published proxy records”?

Line 87 – missing “)”

Line 93 – “Shifting”

Lie 96 – change “are argued to reflect” to “may reflect”
Line 106 – change “those” to “these”
Line 111 – and “a” before Mn-rich
Sentence beginning 113 -break into multiple sentences. “...increases at the glacial/interglacial transitions. At U1314, these increases also occur during interglacial periods. Increased Mn/Ca is indicative of more oxide conditions, ...”
Line 132 – “at” should be “in all oceans”
Sentence beginning 131 should be split – New sentence after [53], “These observations eliminate temperature-driven oxygen...”
Line 135 remove “,” before “or”
Line 177 – either “increases in ventilation” or “increased ventilation”
Sentence beginning line 213 – The timing of the decrease in Mn/Al. and P/Al ratios and reactive phases displayed by the U1314 records is also coeval with the largest decreases in BPWO...”
Line 220 – “used”? should this be “observed” or possibly just removed and read as “compared to U1385”
Line 224 – “which could be related...”
Line 227 – “suggest substantial shifts” or suggest a substantial shift”
Line 228 – remove “anyways”
Line 233 – “reductions in deep-water...”
Line 249 – a word is missing before geochemical
Line 260 “will” should be “would”
Sentence beginning line 323 – should be split into two sentences
Line 333 – “critical data” or “critical evidence” not “critical data evidence”
Line 362 – why not give the actual number of samples?
Line 364 – use approximately here instead of around
Line 368 – I think the samples were diluted to 50 mL rather than diluted with 50 mL of Milli-Q
Line 370 – “in the Department...”
In the methods/elemental geochemistry section, it would be useful to include data about precision and accuracy either in the main text or the supplemental material
Line 377 – I’m not sure that is a valid assumption. Other studies have shown that there are significant differences in provenance on glacial/interglacial time scales.
Line 406 – “being” should be “and the reproducibility of all phases was on average...”
Line 420 – “weighed” not weighted
I like the arrows on Fig. 3 – can there be arrows added to Fig. 2 to help explain what we are supposed to see?
The figure caption for supplementary figure 3 does not include the Site.

Reviewer #2

(Remarks to the Author)

I thank the authors’ thoughtful and thorough responses to my previous comments. The manuscript has been thoroughly revised and substantially improved, especially in clarifying proxy limitations, sharpening the discussion, and better integrating supplementary materials into the main text. I have a few remaining comments that I think could further strengthen the manuscript before it’s ready for publication:

Because the authors compare BPWO changes in both polar regions during the MPT and point to similar patterns, it raises a question: which region do you think played a bigger role in driving the MPT? Was the North Atlantic or Southern Ocean? The authors have expanded the discussion of MIS 23 $\delta^{13}\text{C}$ values and multi-proxy patterns. However, the link to broader MPT climate dynamics feels a bit underplayed. Since MIS 23 seems to buck the trend of reduced BPWO, could it be signaling a partial recovery in deep-water ventilation? Might this interglacial represent some kind of tipping point or a transitional interval within the MPT progression? A bit more speculation or contextualization here could be insightful. Overall, I support publication after these remaining points are addressed. This work provides a meaningful contribution to our understanding of bottom water oxygen variability during the MPT.

Reviewer #4

(Remarks to the Author)

Review of MS submission of Hernández-Almeida et al. entitled “Glacial dysoxia in the deep subpolar North Atlantic during the Mid-Pleistocene Transition”

Hernández-Almeida et al. present an impressive amount of data from IODP Site U1314 from the North Atlantic addressing climate- and carbon cycle dynamics during the MPT. This is a very important long-term climate transition that manifested itself without any notable changes in the external (i.e., orbital) forcing, pointing at the importance of internal climate feedback mechanisms. The central motivation of the study of Hernández-Almeida et al. is to better characterize these internal climate feedback mechanisms through the reconstruction of redox-sensitive proxies in the marine sediment record of IODP U1314, particularly illuminating possible changes in North Atlantic carbon storage variations and how they are linked with northern-hemisphere ice sheet dynamics and Atlantic overturning dynamics. Here I provide my evaluation of the study, joining the group of reviewers after the first revision round. I cannot comment in detail on the comments of the other reviewers, but overall the paper is too speculative in places and the main conclusions are not sufficiently backed up with proxy evidence.

The study is an impressive amount of work, and I would like to congratulate the authors for this study. I do, however, think that the way the main hypotheses and argumentation is laid out and the figures are presented is not fully convincing. Reviewer #3 has many concerns about the mechanistic concepts that the authors suggested, and I must admit that I share these concerns still.

Major comments

One of my biggest concerns of the study is the fact that the time interval of study is limited to 780 – 1180 kyr. This covers the MPT, while many might argue that an important part of the MPT is missing (e.g. MIS 16). The authors argue that carbon cycle changes in the North Atlantic contributed “to the pCO₂ decrease during the MPT” (abstract). However, the authors provide no indication that these observations of bottom water oxygen changes in the North Atlantic persist post 780 kyr before present, which would explain the long-term drop in atmospheric CO₂ during post-MPT glacials (Hönisch et al., 2009; Chalk et al., 2017). In that sense, I find this conclusion of the study too bold, given the absence of data post MPT.

The authors set out to test the “leading hypothesis” of the MPT being driven by deep ocean carbon changes. Please provide references for studies establishing this hypothesis (l. 44-45). I do not find the results of the study novel enough given the uncertainties regarding the mechanisms put forward (no proxy evidence for surface ocean salinity decline or changes in AMOC strength). The findings of the study in terms of bottom water oxygen variations are not very different from the findings of Lear et al. (2016) [ref. 27] and Thomas et al. (2022) [ref. 28]. Of course, it is always beneficial to get a basin-wide view.

I believe the core finding of the study is the agreement between changes in bottom water oxygen and the supply of IRD to the study site, implying a mechanistic link between meltwater supply through decaying northern-hemisphere ice sheets, AMOC geometry and bottom water changes in the North Atlantic. I find these assertions very interesting but speculative at the same time, in particular because direct evidence of changes in surface and/or bottom water salinities at the study site (e.g. seawater δ¹⁸O) or convincing subpolar Atlantic sea ice variations or changes in the locus of NADW formation is not provided. Possibly, Atlantic overturning strength did not decline during post-MPT glacials (as more and more LGM evidence suggests) but it shallowed and U1314 is influenced by southern-sourced water masses. Generally, I think some of the assertions that are the backbone of the study’s outcome are not sufficiently backed up with proxy evidence. Much of the assertion of AMOC and/or NADW changes hinges on the interpretation of the benthic δ¹³C record, but this proxy does not fingerprint regions of change very well and is influenced by multiple factors (including air-sea gas exchange).

Line 145-148: I find particularly speculative. IRD abundances are interpreted to indicate fresher surface waters, which enhanced surface buoyancy and limit the potential for open ocean deep convection. I presume in the North Atlantic. The only evidence is IRD and benthic δ¹³C. Other likely explanations for the benthic δ¹³C signal are disregarded here, such as end-member δ¹³C changes of deep water masses (thermodynamics during air-sea gas exchange) and/or the increased contribution of southern sourced water masses. Linking sea ice variations to the abundance of *N. pachyderma* is too bold in my view (as this has been traditionally considered as a temperature proxy). *N. pachyderma* can live in sea ice, but it is certainly not “ice related”. Any suggestion on changes in deep ocean ventilation as a result of surface ocean meltwater supply from BPWO is too speculative and should be avoided.

The authors acknowledge different ways of interpreting the U1314 benthic δ¹³C record (both as a signal coming from southern-sourced bottom waters and a changes in the geochemical signature of northern-sourced waters masses). They say “we are unable to clearly identify the water mass or process responsible for the decrease in BPWO”, which is bold and honest. Any suggestion that follows between lines 202-205 (and in fact in the Abstract) and in lines 230-234 remains thus a speculation in my view. The rest of the paper, despite this limitation, holds onto the interpretation of “weakening and shallowing of the AMOC upper cell” despite the limitations mentioned above. This is a little bit of a conundrum. Assertions such as in the response to Reviewer #1 (comment 1) that the authors show “that coupling between ocean and cryosphere in this region was instrumental in the slowdown and shoaling of deep-water production in the subpolar North Atlantic” is contradictory to that statement.

Comparison of U1314 with ODP1094: I do think that the comparison to Mn/Ca changes at ODP Site 1094 is quite a stretch. Suggesting that both cores have robust chronologies glosses a little bit over the fact that both records come with age uncertainties, in particular ODP 1094, where the benthic ¹⁸O record has quite some gaps because of poor carbonate preservation. The statement in lines 289-291 are entirely unclear, and require more explanation. I concur with Reviewer #3 that the link to the southern hemisphere is premature and overly ambitious, still despite the revisions. In my view, ODP Site 1094 indicates changes in BWPO in southern-sourced bottom waters, in agreement with Farmer et al. (2019). Any link to the freshwater budget at the surface cannot easily be linked to convection or deep water formation in the Southern Ocean, as this happens through local processes near the Antarctic margin.

Structure of the manuscript: I must admit that I am a bit puzzled by the set up of the manuscript. After the introduction, in the section “Records of past BPWO during the MPT” the reader is referred to quite a lot of supplementary figures and extended data although this seems to be key data of the manuscript. This is confusing, and indicates that the manuscript is not as streamlined as it could/should be. Reviewer #1 has also mentioned difficulties to refer to the data in the figure. This might have improved in the revised manuscript, but referring mostly to Supplementary Material in the first quarter of the Results-Discussion section is counterproductive.

New data: I am a bit confused what data are newly presented. In line 105, it is stated that “previously published Mn/Al ratios in bulk sediment and Mn/Ca ratio in planktonic coatings at those two sites are used as proxies for pore water oxygenation”. Here I would like to see references to the publications in the text, rather than reference to “Supplementary Information”.

Some reviewers mentioned discrepancies between some interglacial stages in the response of the BPWO proxies, and inter-proxy differences. The authors have addressed these but there remains uncertainties in how valid absolute BPWO reconstructions based on these semiquantitative/qualitative proxies are. It shows in some of the proxy responses during some interglacials (not as high as expected) or terminations (still low oxygen despite onset of interglaciation). There is some uncertainty around redox-sensitive proxies that let's one doubt whether these features are true signals or in fact sedimentary artifacts (due to changes in redox-front etc.). The sections on sedimentation changes helps but there could still be changes in sediment composition, sediment porosity that drives the redox-proxies, although I believe the wide-spread similar response in BPWO in the North Atlantic supports the findings.

Minor comments

Many statements in the manuscript are too vague in my view, and need specification. They make it really hard to follow: "benthic foraminifera data" (l. 23), "we analyzed proxies" (l. 28), "previously published proxies" (l. 73), "new and previous proxy records" (l. 82), "previously published datasets of surface and deep oceanography", "Mn minerals" (l. 109), "geochemical data" (l. 141, I don't think IRD does not really count as geochemical data.), "as the other redox proxies" (l. 155), "in certain moments" (l. 160), "as well as with other proxies indicative at Site U1314" (l. 171), "using multiple independent proxies" (l. 195), "deep-ocean circulation proxy" (l. 226; eNd is strictly a water mass provenance indicator), "deep-water formation in convection sites" (l. 232, what convection sites are here specifically referred to?, also in line 260), "deep-circulation records" (l. 249), "lag between benthic organisms in the North Atlantic and changing deep-ocean conditions" (l. 253, how can benthic organisms lag? Do you mean their response?), "AABW filled the AMOC bottom cell" (l. 296, that seems to me the very definition of the lower cell), "enhancing the non-linear response of the climate system" (l. 339, what exactly is referred to here?)

L. 48: it should say stronger interglacial ACC flow.

L. 62: it should say "build up" or "increase in respired carbon in the subpolar North Atlantic reservoir" or similar.

L. 63: I also do not agree with the statement on the "absence of oxygen-sensitive proxies", when studies from the area exist (Lear et al., 2016; Thomas et al., 2022).

L. 77-78: statement is very vague.

L. 90: AMOC lower limb, the NADW

L. 102: 365 $\mu\text{mol/kg}$ for AABW is too high (Figure 1 shows that [O₂] at ODP Site 1094 is much lower).

L. 110: Add reference to sentence that ends in line 110.

L. 126: Add reference to sentence that ends in line 126.

L. 136-138: weird phrasing ("we attribute changes in BPWO [...] to changes in deep-ocean oxygenation"). I see that there the physical/dynamical driver of oxygen is intended to be emphasized, but this should be specified.

L. 145. Add reference or figure callout to sentence ending in line 145

L. 158. Add reference or figure callout to sentence ending in line 158

L. 169. Add reference or figure callout to sentence ending in line 169

L. 187. The reason is unclear why specifically "Rockall Plateau" is important here, which is quite a narrow area in the North Atlantic.

L. 188. Irminger Basin

L. 216: 5-15 $\mu\text{mol/kg}$? I also recommend to mention where IODP Site U1385 is located.

L. 218+222+263+280+282. Add reference and/or figure callout to sentence ending in line 218+222+263+280+282

L. 221. Define Dd13Cb

L. 226. I don't see how eNd can information about the AMOC strength, when this proxy mostly indicates water mass origin

L. 240. "Higher ice mass" (as in tall) or more expanded ice sheets (as in more area)? Please specify.

L. 251 and figure axes labels: "well ventilated benthic foraminiferal assemblages" is a strange phrasing. It is not the foraminifera that are well ventilated, but the bottom water environment that they track. Please rephrase.

L. 301. Unclear why a number of eNd studies are cited when referring to "increasing trends in deep ocean nutrient and dissolved inorganic carbon concentrations"

L. 425. "Foraminiferal identifications were based on specialized literature". Either include citations or remove.

I wonder why the authors haven't applied more quantitative bottom water oxygen assessments based on the BFOI index of benthic foraminiferal assemblages or similar.

Version 2:

Reviewer comments:

Reviewer #2

(Remarks to the Author)

I have no further comments. The authors have addressed all of my previous concerns, and I support publication of this manuscript.

Reviewer #4

(Remarks to the Author)

I thank the authors for the thoughtful and thorough revision of the manuscript and the responses to my comments. I do have remaining reservations against the outlined argumentation in the manuscript, but I believe these are my own personal reservations that go beyond the scientific soundness of the multi-proxy dataset and manuscript – the biggest reservation concerning the time period covered. I do appreciate the inclusion of additional references, comparisons with external data and revisions of the manuscript, which in my view strengthen the arguments made in the manuscript. In light of the positive response by reviewer 1 and 2 and the thorough revisions, I am looking forward to seeing the response of the community to the study. I therefore recommend publication of this study in Nature Communications.

REVIEWER COMMENTS

Dear Editor and reviewers,

Thank you for your careful reading of our manuscript and the constructive suggestions to improve it. Based on these, we attended to four major modifications in this revised version:

- 1) Added additional discussion on how the different proxies compare at Site U1314, highlighting discrepancies and limitations.
- 2) Included additional discussion of the potential mechanisms involved in the BPWO changes.
- 3) Provided support for globally relevant implications and a clearer explanation of bipolar teleconnections during the MPT.
- 4) new synthesis figure (Fig. 4) to summarize main changes before and after MIS 26 (960kyr) and possible scenarios of deep circulation.

A detailed point-to-point replies can be found below in red. All line numbers are lines in the manuscript with tracked changes.

Reviewer #1 (Remarks to the Author)

Hernandez-Almeida et al. have pulled together an interesting set of data to try to understand bottom water and pore water oxygenation and how that relates to deep water circulation. The data is interesting and of high quality. Ultimately, I do think the data and interpretations merit publication, but I think the manuscript requires some significant modifications. In particular, I think that the figures could be constructed in a way that is less confusing.

Response: We thank the reviewer for the opportunity to add clarifying information to the manuscript and figures and have done so in the revision.

For example, some of the data is presented in multiple figures while the reader is forced to compare at times between figures. For example, line 152 references Figs. 1, 2 and supplementary Fig. 4, but each of these figures has multiple parts, and I have no idea which parts I am supposed to look at.

Respective figures panels are now explicitly referenced in the text.

In addition, there are a lot of broad statements about the MIS 24-22 transition, for example there are multiple places where the interval from 940-870 kyr is discussed with respect to several proxies. The authors suggest that the proxies demonstrate that there is low BPWO across this interval, but the benthic foram ventilation proxy does not agree with this statement. Fig. 2D/3B suggests relatively higher BPWO across MIS 23 in contrast to the other proxies that suggest low BPWO across the entire interval from MIS 24-22. The discrepancy between these proxies needs to be explained. All of the geochemical proxies do agree. The difference between the benthic forams and the geochemistry needs more attention.

In the new version of the manuscript, we have now clarified that the most dramatic reductions in BPWO occur during MIS 22 and MIS 24, with a plausible lower BPWO during MIS 23. It is important to highlight that the double peak in the faunal data from Site U1314 vary alongside P/Al and Mn/Al records. This is explained because it is only below a certain threshold that oxygen is expected to be a key driver of assemblage composition and thus causing a noticeable change in the species composition.

Similar discrepancies among proxies are also observed for several other proxies at Site U1314 and in downstream sites. For example, at Site U1385, whereas the O₂ reconstruction during entire MIS 23 shows high BPWO (reconstruction truncated at 235 $\mu\text{mol kg}^{-1}$), the planktonic foraminifera Mn/Ca data indicates only one point with high values (i.e., high oxygenation during MIS 23) (Fig. 3D). Moreover, the benthic $\delta^{13}\text{C}$ data at that site shows a typical “double peak pattern” observed also in the benthic carbon isotope record of other North Atlantic sites (See extended data), with a negative excursion of ca. 0.7‰ at 910 kyr, which is not observed in the O₂ reconstruction at Site U1385, reflecting the different sensitivity of the different proxies to BPWO levels and hinting to potential local influences (e.g., carbon flux intensity from the surface ocean or lateral advection of carbon). We have added some sentences aiming to discuss the discrepancies between these faunal

and geochemical proxies. This information can be found in L. 245-254, which answers also to reviewer #2's comment regarding the need to incorporate the limitations of each proxy

Other comments:

1. Some of the sentences are too complicated. Sentences should include one idea. There are also some awkward phrases, such as "converge in the idea" line 50. **We have revised the manuscript and simplified sentences, such as the one mentioned by the reviewer.**
2. I wouldn't call an authigenic mineral a contaminant - line 102. **We have removed "contaminant" from that sentence and throughout the manuscript.**
3. Some of the colors used in the figures are difficult to differentiate (Fig. 2E and 2F, Fig. 3G). **We have changed the colors of the figures to improve their quality and readability of the figures.**
4. Figure 1 is not easy to read. Thank you for your comment, however, **we are not sure what the problem with the figure is. Is the reviewer referring to how low-resolution this figure is, making much of the labelling pixelated? We have made the font size larger to improve readability of the sites.**
5. Table 1, there is no time indicated for the Na-acetate solution or for the organic step. **We have added the missing information to Table 1: 4h Na-acetate solution and 24h shaking for the organic step.**
6. The way the figures were constructed is a little bit confusing and the reader is asked to move back and forth between fig. 2 and 3 and figures in the supplemental information. **We have kept a more logical order when we refer to the figures and supplementary information in the text.**
7. The reader is referred to the MIS for Fig. 3, but these are not listed on the figure. **Thank you for spotting the missing MIS labels. In the new version of the manuscript, we have added this information to figure 3.**
8. Figure 2E caption needs to include a mention of these taxa being sea-ice related as indicated in the text line 175. Also, the trends in the figure and relationship to other data is not clear. **We have added "sea-ice related plankton species" to the caption, as suggested. Regarding the trends in the figure, we are not sure what the reviewer refers to. Figure 2E shows that when there are high values of these sea-ice related taxa, there is a lower BPWO, as indicated by the multi-proxy evidence, and the negative correlation shown in Supplementary figure 6. We have added "where high percentages of this species correspond to low BPWO at Site U1314" in the caption and arrows so the reader has a quick overview which records relate to O₂ and which to other environmental conditions. (similarly to Fig. 3).**
9. From the graphs, I don't see evidence of low BPWO in MIS 21 or 25. **The reviewer is right in his/her appreciation, proxies do not indicate low BPWO in MIS 21 and 25. We intended to indicate that during the onset of interglacial phases after MIS 25 and MIS 21 BPWO is still low during (L. 256). We have modified the text to avoid any misunderstanding.**
10. It is very confusing to refer to MIS when that information is not available for all figures. Sorry for the confusion. **The MIS has been added to the upper part of all the figures.**
11. There are numerous places where ", and" occurs when the "," is not needed. Thank you for noticing this. **The text has been revised and modified.**
12. I think the method is actually Method 3050, not 305 (line 267). **Thank you for spotting this typo, we have corrected the sentence.**

13. I gram of sediment in a 15 mL centrifuge tube seems like a lot (line 299). 0.1 g was used for the total digestion, should 0.1 g also be used for the sequential P extraction? **Yes, it was a typo, we used 0.1 g for the sequential P extraction.**
14. Instead of saying Mn spikes, perhaps maxima? (Line 15 of supplemental information and elsewhere). **We have changed Mn spikes to Mn maxima.**
15. By phosphorus provenance (line 70 supplemental information) do you mean the different pools of P? This is a confusing use of provenance. Presumably reactive P was once related to organic matter. **The reviewer is right. Consequently, we have replaced by “different phosphorous pools in the ocean”.**
16. Sorry these last comments are out of order. Line 41 states that the deep ocean is the largest C reservoir, but it actually sedimentary rocks. **We have completed the sentence by writing that the “ocean is the largest carbon reservoir that readily exchanges with the atmosphere”.**
17. Line 77 - it would be helpful to identify the two sites downstream by site number. The two **sites downstream have been added in (U1314, 1094).**

Reviewer #2 (Remarks to the Author):

This manuscript provides an insightful analysis of bottom water oxygenation (BPWO) changes across the subpolar North Atlantic during the mid-Pleistocene transition (MPT), emphasizing the role of ice sheet dynamics, freshwater input, and Atlantic Meridional Overturning Circulation (AMOC). The authors reconstruct BPWO using multiple proxies—including solid-phase manganese (Mn) enrichments in bulk sediments and planktonic foraminifera coatings, sedimentary phosphorus (P) reservoirs, and benthic foraminiferal assemblages—at IODP Site U1314. By integrating previously published records, the study highlights the role of cryosphere-ocean coupling in the North Atlantic and Southern Ocean in controlling ventilation, BPWO, and carbon storage, potentially offering new insights into the mechanisms of the MPT.

However, several issues should be addressed to improve the manuscript. While the use of multiple proxies is a strength, the technical details require further clarification, and the discussion could better incorporate the limitations of each proxy.

We thank the reviewer for the opportunity to add clarifying information to the limitations of each proxy. Consequently, we have added information on the limitations of each proxy in the methods section.

Additionally, the structure and flow of certain sections, particularly the methods and supplementary information, could be improved for clarity. Most importantly, the discussion of the mechanisms needs further clarification.

Major comments/suggestions are as follows:

1. Introduction:

While the introduction effectively synthesizes previous studies, it should better highlight the originality and purpose of this study. The research gap addressed by the authors should be explicitly stated.

This is a fair point, and we are now paying closer attention to the originality of this study. The revised text can be found in L.62-65. In a few lines, we emphasize that unlike previous studies, which hypothesize that processes occurring in the Southern Ocean was instrumental in starting oceanographic and climate changes that resulted in an increase in C storage in the deep ocean during the MPT. In our study we show, using a multi-proxy record of BPWO in the subpolar North Atlantic, that coupling between ocean and cryosphere in this region was instrumental in the slowdown and shoaling of deep-water production in the subpolar North Atlantic. This enhanced

storage of respired CO₂ in that region and highlights the importance of studying and understanding local and/or regional processes when interpreting O₂ related proxy data.

2. Methodology:

The use of multiple proxies is a strength of this study; however, the methodological details require further clarification. Clearly distinguish which proxies were measured by the authors and which were obtained from previous studies.

For this study, we measured bulk Mn/Al and P/Al, sequential P extraction, total C and analyzed benthic foraminifera assemblages. Description of methods used to generate these measurements are stated in L. 891-946. At L. 881, we indicate that the benthic foraminifera stable isotope data from Site U1314 was previously published. We have added that Mn/Ca in planktonic foraminifera coatings from Site U1314 and 1094 were obtained from previous studies in L. 157.

Please provide a more detailed description of proxy and potential diagenetic effects to improve transparency. For example, clarify how Mn/Al and P/Al were validated against independent redox indicators, and the potential post-depositional alterations of P phases and their impact on interpretations. Both the methods and Supplementary information need restructuring.

We are not sure what rev#2 exactly asks for by “validation against independent redox indicators”. Since we are using a multiproxy approach combining geochemical and faunal derived proxies, we are providing independent validation for either type of proxy data. Furthermore, all proxy data types are well established (see the most recent review by Hoogakker et al. 2025) and used as redox indicators in paleoceanographic studies already for decades. Correlation among the proxies sensitive to BPWO and other surface and deep ocean conditions at Site U1314 can be found in Supplementary Figure 6.

We have revised and reorganized the methods and supplementary information to be able to include this information on the potential diagenetic effect on geochemical proxies, and added the following information:

Supplementary information 30-37: “A shallower oxygen penetration depth would shift the manganese (Mn) peak into more reducing pore waters, where Mn would dissolve and diffuse upwards, effectively removing any trace of the former peak. In contrast, if the oxygen penetration depth increases in the sediment, the Mn peak remains in oxygenated conditions and is thus preserved (Mangini et al., 1990). One advantage of using Mn/Ca ratios in foraminiferal calcite over bulk sediment proxies like Mn/Al is that, once Mn is incorporated into the foraminiferal shell, its concentration remains stable and is not affected by subsequent diagenetic redox processes (Koho et al., 2015; McKay et al., 2015).”

Supplementary information 90-103: “Phosphorus can be remobilized in sediments after deposition. The main consequence of diagenetic alterations of P involve redistribution between phases (sink switching; Ruttenger and Berner 1993). Once buried below the most reactive surface layers, the bulk P concentrations are largely locked in, although redistribution occurs of P phases cause a decrease in the fraction of organically-bound and surface-bound P and an increase in authigenically-bound P in the form of authigenic carbonate fluorapatite minerals with increasing age/depth (Filippelli and Delaney, 1996). However, there are no apparent trends with age/depth in the authigenic P fraction at Site U1314 (Supplementary figure 3). Additionally, if diagenesis did redistribute P from the organic or Fe-bound P into the authigenic phase, this would not affect to our interpretation since the total amount of originally reactive P would still be locked into the sedimentary record at Site U1314 that we are interpreting.

3. Use of Mn Proxies for BPWO: The text describes how Mn/Al and Mn/Ca serve as proxies for bottom and porewater oxygenation. However, it would be useful to clarify how these proxies compare with other commonly used redox-sensitive elements (e.g., U, Mo, V). Are there any limitations or uncertainties associated with Mn-based proxies in these records?

Unfortunately, there are no records of other redox sensitive elements at Site sU1314 and 1094 that can be used for comparison with our set of proxies. The proxies used in our study are widely used, as evidenced by their inclusion in the latest BPWO proxy review by Hoogakker et al. (2025).

Limitations and uncertainties of Mn/Al and Mn/Ca are discussed in the Supplementary information. We consider that there is not a single proxy that can be considered as better than another redox proxy. Of course, all proxies have their unique strengths and limitations, including U, Mo, and V, and their use depends on the study settings (Tribouillard et al 2006). This is relevant particularly for U, because U redox conditions can be potentially linked to organic matter burial (McManus et al 2005). The safest approach it is to integrate different proxies, which enhances the robustness of the conclusions, this approach is acknowledged by reviewer #1 in point #7 of his/her review.

The following are examples of studies that compare one or several of the proxies used in our study with other commonly used redox sensitive in the same core :

- 1) Rohde et al. (2021) compared Mn, Re, and U measurements during MIS 11 at Site 1094, and found evidence of consistent behavior of these elements with porewater oxygenation variation.
- 2) Boiteau et al. (2012) documented a pattern of elevated Mn/Ca ratios during interglacial intervals that correspond to low U/Ca levels.
- 3) Thomas et al. (2022) observed a lower P/Al ratio at Site U1308 that is interpreted as a proxy for BPWO, and the PF Mn/Ca and benthic foraminifera carbon gradient at Site U1385, show good agreement supporting the interpretation provided by the P/Al ratio.

4. Proxy interpretations: Most proxies indicate lower BPWO during the MPT; however, discrepancies among them need further explanation. For example: while Mn/Al suggests generally lower BPWO during the MPT (Fig. 2B), well-ventilated foram% only show lower abundance during MIS 24 and 22 but higher abundance during MIS 23 (Fig. 2D). The reasons behind these inconsistencies should be explicitly addressed. Consider alternative interpretations and discuss the robustness of each proxy.

In the revised version of the manuscript, we have now clarified that the most dramatic reductions in BPWO occur during MIS 22 and MIS 24, with a plausible decrease in BPWO during MIS 23, as inferred from the lower values of Mn/Al and P/Al. Whereas the faunal data from Site U1314 show a similar pattern, the values do not suggest lower BPWO, except for a short episode at 910 kyr, which coincides with lower benthic $d^{13}C$. We argue that this difference in the proxy data is due to the different levels of proxy sensitivity; only below a certain threshold is oxygen expected to impact/change the benthic foraminifera assemblage composition. Although species adapted to high oxygen levels may survive in dysoxic environments, their relative abundance only decreases when oxygen becomes a limiting factor and competition with other species decreases their abundance (Van der Zwaan et al., 1999; Moffit et al 2015).

Similar discrepancies among proxies are also observed in several proxies at Site U1314 and in downstream sites (O_2 reconstructions from Site U1385; Thomas et al. 2022). At Site U1385, the O_2 reconstruction shows high BPWO (reconstruction truncated at $235 \mu\text{mol kg}^{-1}$) during the entire MIS 23, whereas the planktonic foraminifera Mn/Ca data reveal only one point with high values (indicating high oxygenation during MIS 23). Moreover, the benthic $d^{13}C$ data at that site shows a typical “double peak pattern” observed also in the benthic carbon isotope record of other North Atlantic sites (see extended data), which is not observed in the O_2 reconstruction at Site U1385, reflecting the different sensitivity of BPWO proxies to oxygenation levels.

Following reviewer suggestion, we have added a discussion about discrepancies between these faunal and geochemical proxies, which can be found in L. 245-254. This change also accommodates other reviewers comment suggesting the need to incorporate the limitations of each proxy.

5. Mechanisms: The authors argue that BPWO in the North Atlantic and Southern Ocean was primarily controlled by the ventilation, but other potential influencing factors (e.g., temperature-oxygen solubility and productivity-driven oxygen consumption) should be addressed. If ventilation was the primary control on BPWO, clarify why North Atlantic ventilation was specifically driven by freshwater forcing from iceberg melting. Further elaborate on how these processes relate to the broader climatic transition during the MPT. Expand the discussion on the role of Southern Ocean processes and reference additional studies where relevant specifically driven by freshwater forcing

from iceberg melting. Further elaborate on how these processes relate to the broader climatic transition during the MPT.

That is an interesting reflection. Productivity-driven oxygen consumption has been ruled out as the dominant process driving BPWO during the coldest intervals at Site U1314, since high productivity occurs only during interglacials (Hernández-Almeida et al. 2012; 2013), when highest inferred BPWO is observed. Temperature-oxygen solubility is not driving BPWO changes since the MPT is characterized by deep-ocean cooling, and with temperatures relatively stable (Bates et al. 2014). We have added this information to the discussion (L. 219-222). Following the reviewer's suggestion, we have elaborated further on how the meltwater forcing leads to changes in the surface (e.g. via cooling and sea formation) and deep ocean (circulation and BPWO). We have added a new supplementary figure 7 showing cross-correlation between proxies to explain the chain of events since the IRD delivery (L. 435-444). We also strengthen our discussion on the role of the Southern Ocean processes and how teleconnections might delay the resumption of NADW production via freshwater forcing from ice melt. This new discussion can be found in L. 435-444.

6. Conclusion: The conclusion effectively summarizes the findings but should more explicitly discuss the broader implications for MPT climate dynamics.

Thank you very much for pointing this out. We prefer to remain cautious regarding broader implications for MPT climate dynamics, because the lack of additional proxies, e.g. high-resolution pCO₂ records (either from ice or marine cores), prevents linking our findings with MPT climate dynamics. Our attempt to discuss broader implications of our findings with the MPT climate dynamics has been criticized by rev#3 (see his/her last comment), consequently we prefer to avoid speculate further.

Additionally, consider explicitly stating whether the findings align with or challenge previous models of deep ocean carbon sequestration during the MPT.

We have now specified how our findings fit these previous models. Please refer to L. 599-647 and references included in that paragraph.

Consider discussing BPWO changes before and after the MPT to provide additional insights into the mechanisms driving the transition.

Unfortunately, our record covers only the interval from MIS 31 to MIS 19; so, it is not possible to add additional insights into the mechanisms operating before and after the MPT. However, since rev#3 comments on the glacial intervals, namely MIS 30 and MIS 22, having also low BPWO, we are now providing additional insights about them, and how similar BPWO reductions are observed in downstream records (L. 340-397).

7. The section on BPWO during the MPT provides a well-structured discussion using proxy records from IODP Site U1314 and ODP Site 1094. The integration of different geochemical proxies (Mn/Al, Mn/Ca, and phosphorus phases) is commendable, as it enhances the robustness of the interpretations regarding bottom water oxygenation. Consider briefly discussing why these sites were chosen specifically to study BPWO during the MPT, especially in the context of past studies. More explicitly compare BPWO trends between sites U1314 and U1385 is needed, particularly in relation to $\delta^{13}\text{C}_b$ signals.

These sites were chosen because i) they are located downstream of newly formed deep-water masses in both hemispheres, hence providing evidence of changes in BPWO; ii) they have continuous and high sedimentation rates over the MPT; iii) several relevant proxy data to reconstruct changes in surface and deep hydrography are available and iv) their chronology is robust. This information is detailed in L. 159-161.

Comparison of $\delta^{13}\text{C}_b$ between U1385 and U1314 is provided in the Extended data and also at L. 340-400 in the main text.

8. Figures should be explicitly referenced in the text. When citing figures, specify which curve is being discussed (e.g., Fig. 3A rather than simply Fig. 3).

Figures panels are now explicitly referenced in the text.

Additionally, consider adding a figure synthesizing the hypothesized sequence of events connecting freshwater forcing, ventilation changes, and carbon sequestration.

The new figure 4 synthesizing the sequence of events is now added to the manuscript (L. 644).

Also, supplementary references should be more clearly tied to the discussion. Instead of simply citing them, briefly indicate what they illustrate and how they support the argument.

Thank you for pointing this out. We have made a better integration of supplementary figures in the main discussion.

More specific comments:

Line 60: Specify which hypothesis is being referenced?

Rev#1 made a similar remark and suggestion. Following both reviewers comments, specific indications about the hypothesis to be tested and the research gap that have been addressed has been added to L. 91-95.

Line 77: Clarify which two sites are being referred to. Line 80: Provide a reference?

Sites added in brackets (U1314, 1094) at L. 157, and reference added.

Line 86-89: Expand on how the interpretation of lower $\delta^{13}\text{C}_b$ at Site U1314 distinguishes between reduced NADW ventilation and intrusion of southern-sourced waters. Are there additional proxies (e.g., Nd isotopes, benthic foraminiferal assemblages) that support one interpretation over the other?

Unfortunately, based on the $\delta^{13}\text{C}_b$ record at Site U1314 and on the other proxies we are not able to distinguish unambiguously between reduction in NADW ventilation or an intrusion of southern-sourced waters. Instead, we suggest that the weakening of NADW production and shoaling of deep-waters formed in the high-latitude North Atlantic enabled the expansion of southern-sourced waters, with higher capacity to store C in the deep ocean. The multi-proxy evidence at Site U1314 would be compatible with a combination of these processes.

Explain the higher $\delta^{13}\text{C}_b$ values during MIS 23.

We have added additional explanation about the more positive $\delta^{13}\text{C}_b$ values during MIS 23 at L. 245-254. A more detailed explanation is provided below. However, providing a detailed explanation is beyond the scope of the current manuscript and also not possible due to the number of words limitation.

Positive $\delta^{13}\text{C}_b$ values during MIS 23 is a common feature in all sites across the Atlantic basin (e.g., Raymo et al., 2004), but the maximum is of lower magnitude than in preceding and subsequent interglacials (see Extended data), whereby the MIS 25 $\delta^{13}\text{C}_b$ maximum, similar to MIS 13, coincides with an eccentricity minimum (for detailed discussion on links between $\delta^{13}\text{C}$ and eccentricity see Wang et al. 2003 and 2010). Moreover, all records also include several negative excursions (with different magnitude depending on the depth/latitude), which are related to abrupt climate deterioration events and associated reduction in deep-water formation, leading to lower BPWO according to the Mn/Al and P/Al at Site U1314, and planktonic foraminifera Mn/Ca at Site U1385. Overall, the evidence suggests that NADW production occurred in certain moments, when freshwater input to the high-latitude North Atlantic was at its minimum (according to the lowest IRD/g values between 915-911 kyr and between 900-890 kyr), and sea ice cover was absent/minimal in the higher mid-latitudinal North Atlantic (as indicated by decreases of 50-40% in the ice related microfossils *N. pachyderma* and *C. davisiana*).

Line 119-122: Specify which productivity proxies were analyzed and explain why productivity changes did not drive oxygenation changes.

Thank you for the comment. In the new version of the manuscript. We specified that we used total organic carbon, opal and calcium carbonate as primary productivity and export productivity proxies, and added an explanation about why we consider that these factors are unlikely driving bottom water oxygenation changes (L. 216-226).

Line 155-158: Justify why BPWO changes are attributable to ice sheet dynamics rather than other factors.

Sorry for the confusion, which is mainly a language issue. Our argument is not that BPWO is attributable to ice sheet dynamics per se, but is a consequence of the interaction between ice-sheet instability (through iceberg and freshwater delivery, which changes surface ocean buoyancy) and ocean dynamics (through reduction in deep-water formation in the subpolar North Atlantic) leading to a reorganisation in deep ocean circulation with cascading impacts on the BPWO. We have modified the text in L. 234-237 to clarify this point.

Lines 143-157: The section discussing "A less active deep-water convection..." should better distinguish between local and basin-wide processes.

We have thoroughly revised this section and differentiated between local processes (i.e. changes in hydrography in the subpolar North Atlantic, as evidenced by Site U1314 and other neighboring sites) (L. 234-2691) and basin-wide changes, which include downstream sites U1308, 607 and U1385 with proxy records sensitive to BPWO, and others sites such as 1267 and U1479, which show changes in ocean circulation and BPWO in agreement with evidence at Site U1314 (L. 297-410).

Line 274: The dataset includes bulk elements such as P, Ca, Al, Ba, Ti, Fe, Mg, but only shows P, Mn, and Al. Consider providing all measured elemental data. Did the authors measure U concentrations? As a redox-sensitive element, U could provide additional insights into oxygenation changes.

We have deleted from the methods elements which are not used/shown in this manuscript. Unfortunately, no, we did not measure U concentrations.

Line 326: Address how samples with no well-ventilated benthic foraminifera were handled. Was this based on a minimum count of 200 individuals? Could absence be due to random sampling bias?

The assemblage count was consistently based on at least 200 individuals. This information is provided in the methods (L. 884), and the exact total number of individuals counted per sample is listed in the datasets that are available in PANGAEA. We counted the full assemblages, comprising 64 species. Because of the high dimensionality of this dataset, and the potential environmental factors that may drive some of the species not related to ocean ventilation (e.g. amount and type of food, temperature, alkalinity, etc.), we decided to discuss only the species potentially indicative of well-ventilated conditions. The sampling for benthic foraminifera analyses was not random, but done at a consistent 8 cm spacing (L. 877). The full census counts can be made accessible to the reviewers, but since the rest of the species are not discussed here and will be the focus of a future manuscript by co-author P. Diz, they are not included in the datasets archived in PANGAEA.

Figures 1 and 2 are referenced, but their role in supporting key arguments could be more explicitly stated in the text. For example, how does Figure 2A specifically demonstrate reduced ventilation or changes in nutrient concentrations?

Supporting key arguments when figures are referenced have been added to the main text (e.g. at L. 161).

Supplementary information

References to figures and supplementary information should be more explicitly tied to the discussion. Instead of simply citing them, the authors could briefly indicate what these figures illustrate and how they support the argument.

Thank you for your comment, a more direct link between figures and supplementary information and the discussion has been made throughout the manuscript.

Line 10: "Under oxygen-rich conditions, Mn is then removed from porewater..."—the wording suggests an absolute removal process. Consider specifying that Mn precipitation as Mn-rich carbonates or oxides is a dominant but not exclusive removal mechanism.

Wording has been modified following the reviewer's suggestion (L. 26).

Line 34-37: The observed 5–7 kyr offset in Mn peak timing between Site U1314 and Site 1094 is attributed to burial conditions or age model uncertainties. Could other mechanisms, such as regional differences in deep water ventilation, contribute to this offset? This point should be explicitly considered.

Thank you for your comment. We do not attribute the offset to chronological issues. Actually, we discard any age model issues because both cores are aligned to the LR04 benthic stack. Instead, we consider other processes which are indicated at L. 727-734 in the main text. We argue that the magnitude of initial freshening in the convection areas of the Northern Atlantic (based on the larger IRD peaks at Site U1314 during MIS 26), as well as different rates of deglacial warming in the deep Atlantic and in the Pacific, with midsummer insolation at 65 °S increasing 10 kyr prior to its NH counterpart (Schulz and Zeebe, 2006), are factors that could be responsible of the delayed in reoxygenation of the deep ocean in the high-latitude North Atlantic.

Line 36-39: The timing of CO₂ release relative to the onset of well-oxygenated bottom-water conditions requires clarification. Are there any leads or lags between these events? Explicitly addressing this question would strengthen the discussion.

It would be very speculative from our side to discuss leads and lags between higher BPWO and CO₂ release because there are not currently available independent CO₂ time series for the MPT. Ventilation of the deep-sea and CO₂ release depend on the overturning conditions in water column. Reduction of freshwater layer or sea ice would promote air-sea gas exchange, which allows O₂ injection to the deep ocean which would be accompanied by a CO₂ release by the upwelling of waters from the deep ocean.

However, cross correlation analyses between independent BPWO proxies ($\epsilon^{205}\text{Tl}$) from the Arabian Sea and pCO₂ over the last deglaciation by Wang et al. (2024) show a strong correlation, which suggests that pCO₂ changes were closely linked with deep ocean ventilation, that simultaneously might have also exerted an essential control over the global ocean oxygen budget. Based on this observation, we could hypothesize that similar near-simultaneous changes in CO₂ release and increase in BPWO could have occurred during the MPT, although we consider that speculating this, in absence of data or analyses to support it, would not strengthen our study.

Line 44-46: The conclusion that the Mn peak offset represents a real delay in deep ocean ventilation re-initiation is reasonable but not fully supported. The discussion would benefit from a more cautious wording or additional evidence.

We have added some cautionary words to this section. Please refer at L. 79-80.

Line 51-55: The difference in Mn/Ca peaks between Sites 1094 and 1090 is linked to burial conditions and initial Mn incorporation into foraminiferal shells. Consider clarifying whether these factors affect only peak amplitude or also timing.

There is no Mn/Ca data for 1090, we assume that the reviewer is referring to U1314. The factors would affect the amplitude, not the time, and this is specified at L. 89.

Line 53-55: How much initial Mn precipitated in the foram shell is needed to create this difference? Please quantify if possible.

Unfortunately, it is not possible to quantify initial Mn precipitated in the foraminifera shells since all the planktonic foraminifera samples were cleaned before measuring trace metals.

Line 56-60: The discussion of Mn contamination would benefit from specifying whether Mn/Ca > 100-200 $\mu\text{mol/mol}$ universally indicates Mn-Fe coatings, or if threshold values vary by site and cleaning protocol. That threshold is specific for Site 1094 but, more general studies including

samples from more regions indicate that values of up to $\sim 200 \mu\text{mol/mol}$ appears to be related to diagenetic Mn-carbonate coatings on the foraminiferal shells (Yu et al. 2007). This has been added at L. 33-35.

Line 72-74: It would be helpful to specify whether these lower-productivity sites also differ in terms of redox conditions, sedimentation rates, or detrital input, as these factors can influence phosphorus burial and remobilization.

Site U1314 and 1094 have comparable sedimentation rates for the MPT of ca. 9 cm/kyr range (Hernandez-Almeida et al 2012; Channell and Stoner (2002)).

For the detrital input, we have compared Ca/Fe values from 1094 (XRF, Jaccard et al 2013) and U1314 (sediment digestion, this study). We can observe that except the short-lived peaks during the interglacial maxima at Site 1094, the values are similar. Moreover, detrital input, as expressed in lithic grains per gram, is very similar in both sites, with values ranging between $2\text{--}8 \times 10^3$ grains/g (Kanfoush et al. 2002; Hernandez-Almeida et al. 2012). Since sedimentation and detrital input in both sites are very similar, we do not think that these factors are influencing burial factors and/or remobilization that could lead to different results in diagenesis conditions. We have added this information to the supplementary material. Please, refer to the new section of the supplementary material "*Sedimentary environments at Site U1314 and 1094*" for detailed explanation and to the new Supplementary figure 1, which is also included below.

Line 85-87: The authors should elaborate on why this mechanism does not apply to Site U1314. Do sedimentation rates play a role? Is there an alternative explanation?

The explanation on why high export productivity does not affect P remobilization due to high oxygen consumption was in the next sentence (L. 134-143): "...this is unlikely at Site U1314, since episodes of higher productivity are characterized by higher burial of reactive P (Supplementary Fig. 3E)."

Line 86: 'episodes of higher productivity', specify the time intervals that correspond to these episodes for clarity

This has been specified in the text, indicating that the higher productivity occurs during interglacial intervals, from which the MIS 25, 21 and 19 show the highest primary and export productivity. Please, refer to L. 131-132

Line 89-105: Instead of using general terms like "redox proxies" or "ocean productivity proxies," the authors should specify which proxies were employed in this study. Additionally, clarify whether the analysis applies to all glacial periods between 1100–780 kyr or specifically to the MPT.

Thank you for the suggestion. We have clarified in the text to which redox and productivity proxies we refer to. In our study, we always refer to the glacial intervals between 1100–780 kyr.

Line 95-99: The comparison between Site 1314 and other locations is useful, but the discussion would benefit from a clearer synthesis of regional differences. Are there factors unique to Site 1314 that influence its productivity-redox relationship differently from these other sites?

There are different patterns in G-IG changes in export productivity, as described in the text, but these do not affect the presence of the Mn peaks, meaning that these Mn peaks are mainly related to changes in the deep ocean oxygenation driven by changes in BPWO and ocean circulation not by higher remineralization of the "local" export production. Please, refer to the new section of the supplementary material "*Sedimentary environments at Site U1314 and 1094*" which explains that there are no factors unique to Site U1314 that influence its productivity-redox relationships.

Line 109-110: During the MPT, $\delta^{13}\text{C}$ records show higher values during MIS 23. How do the authors interpret this? Does it correspond to stronger ventilation?

See our response above regarding MIS 23.

Line 110-112: $\delta^{13}\text{C}_b$ is presented as a deep ocean circulation proxy but is then described as sensitive to multiple factors. Consider rephrasing to emphasize that $\delta^{13}\text{C}_b$ is a complex but valuable proxy rather than implying it is unreliable.

This statement has been modified to reflect the valuable paleoceanographic information from $\delta^{13}\text{C}_{\text{benthic}}$. We have added the following text (L. 181–186): " $\delta^{13}\text{C}$ of *Lobatula wuellerstorfi* has been widely used as a tracer of past changes in deep ocean ventilation. However, the $\delta^{13}\text{C}_b$ signal may be affected by additionally environmental processes, such as air-sea gas exchange (Lynch-Stieglitz et al., 1995), non-stationary water-mass end-members (Charles and Fairbanks, 1990), carbonate ion concentrations (Spero et al., 1997), biological export productivity signals (Mackensen et al 1993) and bias towards specific microenvironment or changes in microhabitat (Gottschalk et al. 2016)."

Line 113-116: The significant correlation between $\delta^{13}\text{C}_b$ and geochemical proxies is an important point. If possible, briefly state which proxies show the strongest correlation.

We have added in the main text (L. 188–190) that the highest correlation occurs between relative abundance of well-ventilated benthic foraminifera and the redox-sensitive geochemical proxies.

Line 118-135: The distinction between *Astrononion novozealandicum* and *A. echolsi* is well-explained, but the connection to the broader study is unclear. Does taxonomic misidentification affect any interpretations in this work?

No, taxonomic misidentification is not relevant for this study. We just wanted to highlight that this species has been identified in the subpolar North Atlantic, in well ventilated environments, but it has been likely named differently. We wanted to include this information because we believe that a correct identification of this species will provide valuable information in future paleoceanographic studies.

Reviewer #3 (Remarks to the Author):

Hernandez-Almeida and co-authors investigated the time interval between 780-1056ka - end of MIS31 into MIS19 covering parts of the so-called Mid-Pleistocene Transition - using site U1313 from the subpolar N. Atlantic. They argue that the time interval covering glacial MIS 22 and 24 was a

period of special interest because it exhibits very low bottom water $\delta^{13}\text{C}$ in benthic forams. These low values apparently reflect dysoxic conditions caused by glacier induced meltwater which then had further implications for the development of the MPT. The rather low epibenthic $\delta^{13}\text{C}$ during the interval in question has been noted before (e.g., Thomas et al 2022), but without further investigating the causes. Thus, the authors provide a multiproxy study to tackle the problem which in my view does deserve appreciation. However, there are a number of major problems with their argumentations. There is a need to overcome these to make this study publishable.

We thank the reviewer for the opportunity to add clarifying information to the manuscript and figures.

The study hinges very much on the rather “traditional” assumption that the glacial-interglacial variability of $\delta^{13}\text{C}$ in epifaunal cibs in the North Atlantic reflects the degree of vertical ventilation (bottom water oxygenation or whatever one likes to call it), i.e., the intensity of the AMOC which is dependent on the production of deep water in polar areas such as the Nordic- Labrador seas. But studies showed a strong habitat-related dependency of $\delta^{13}\text{C}$ -cibs to organic carbon availability in the ambient water, so-called Mackenson-effect.

It is true that we use $\delta^{13}\text{C}$ in epibenthic foraminifera as an indicator of vertical ventilation because it is already available at U1314 and in multiple sites across the Atlantic. It is, however, not the main proxy used for tracking changes in bottom ocean oxygenation and mainly provides a deep-oceanographic context. Our interpretation of bottom water oxygenation relies on proxies sensitive to redox conditions. The maximum $\delta^{13}\text{C}_b$ offset of the Mackensen effect in epifaunal foraminifera would be of 0.4 ‰ (Diz et al. 2007; Mackensen et al 1993; Eberwein and Mackensen 2006), which does not explain the larger amplitude carbon isotope depletions during MIS 24 and 22 (1 ‰). Moreover, these negative excursions in the benthic carbon record occur during episodes of low productivity.

More recently it became clear that even during times of massive meltwater at the ocean surface (eg, HS1) or times during glacial maxima (eg, LGM) AMOC did not stop but rather continued to function (e.g., Repschläger et al. 2021, Howe et al. 2016). Moreover, in a glacial environmental setting of the subpolar to polar northern regions both planktonic and benthic foram $\delta^{13}\text{C}$ signatures are usually strongly depleted during times of enhanced meltwater releases. This would imply that (a) these regions have likely contributed to the $\delta^{13}\text{C}$ signature found downstream. And (b) that southern AABW with its depleted $\delta^{13}\text{C}$ signature – often invoked as evidence for a northward migration during glacials – seems significant only at very large water depths. As we can see in Fig. 1, the investigated cores 1314 (and others, eg, 1313) still belong to the shallower water depth realm.

We agree that the associated deep water mass configuration and transport in the deep Atlantic Ocean during glacial intervals remain highly debated (Muglia and Schmittner, 2021). The studies cited by rev#3 argue for a continued deep-water ventilation in the North Atlantic during glacial intervals, although they offer interesting insights into sources of deep-waters in the Atlantic for their time-scales. However, the extrapolation of their findings to the MPT, and more specifically to the explanation of the data of Site U1314, might appear limited. In Repschläger et al. (2021), the authors found heavy strong contributions of ^{18}O -depleted and ^{13}C -enriched deep waters in the eastern North Atlantic during HS1, a counterpart to the abrupt cold climate events of the MPT glacials and deglaciations, as a fingerprint of northern sourced deep waters formed during HS1, formed by brine rejection (among other mechanisms proposed by these authors). However, at Site U1314 as well as at other sites in the high-latitude North Atlantic (Raymo et al. 2004), we do not find evidence of ^{18}O -depleted and ^{13}C -enriched during glacial, or during IRD-rich intervals (abrupt cold climate events), across the MPT which could indicate an active glacial NADW formation via brine rejection. Howe et al. (2016), combines benthic $\delta^{13}\text{C}$ and Nd to deconvolve of changes in water mass sourcing and nutrient regeneration in the Atlantic during the LGM. However, the records included in this study span from 46° S to 40° N in the Atlantic Ocean, so there is not an analogue as far north as U1314 (58° N) which could be used to test the hypothesis of a glacial NADW filling up depths below 3000 m in the subpolar North Atlantic. Recent work by Blaser et al (2025) proposes

that NADW production was active during the HS1 and LGM and that shifts in benthic isotope data (oxygen and carbon) are attributed primarily to changes in the internal composition of NADW and Southern Ocean sourced waters, and thus do not require major reorganizations of Atlantic source water provenance. According to this study and the mixing models, the $\delta^{13}\text{C}_b$ values of potential waters formed in high-latitudes North Atlantic with seasonal sea cover (and limited air-sea gas exchange) and/or increased organic matter remineralization is close to that of southern source waters, and hence additional proxies such as ϵNd or $[\text{CO}_3^{2-}]$ are necessary for accurate water mass mixing estimates.

Regarding the depth of Site U1314 relative to the boundary between northern and southern source waters, previous studies show evidence that the depth of site U1314 is in the range of the postulated boundary between NADW and SSW during the LGM (previous estimates for the glacial boundary, when the boundary between both water masses shoaled from $>3,000$ m up to $2,000$ m; Curry & Oppo, 2005; Lynch-Stieglitz et al., 2007), whereas Site U1313 and its companion Site 607 are clearly below that depth range and thus strongly influenced by AABW/SSW during glacial periods and periods with a reduced AMOC (Voelker et al., 2010; Raymo et al., 2004). An increased contribution of southern-sourced waters to the Atlantic (rather than changing north Atlantic ventilation patterns) during this interim state is further suggested by the stability of the vertical $\delta^{13}\text{C}$ profiles in the north Atlantic which span the broad window of 1.8 to 0.6 Myr (Raymo et al., 2004). $\delta^{13}\text{C}_b$ at U1313/607 values are just -0.2 per mil lighter than at Site U1314 during e.g. MIS 20 and MIS 22, meaning that both sites were bathed by deep waters with similar composition/source.

Our main conclusion is that a greater amount of respired carbon was stored in the deep North Atlantic, even at subpolar latitudes, during the MPT. We infer that this was achieved by a sluggish deep overturning cell, accompanied by a greater contribution of southern-sourced waters to the deep Atlantic and/or decreased export of NADW. Although the current dataset available at Site U1314 does not allow an unambiguous identification of which water mass was responsible of decrease in BPWO, the lower Nd isotope values during MIS 24 and MIS 22 at mid-latitude sites in the north at sites U1313/607 (Kim et al. 2021; Yehudai et al., 2021) and 1058 (Tachikawa et al. 2021) and south Atlantic sites 1267 (Farmer et al. 2019) and U1479 (Hines et al. 2024) point out at a higher influence of southern source waters. We can not discard the possibility of an active NADW during glacials at U1314 as suggested by the studies led by Howe, Repschläger and Blaser for the LGM and HS1. Whatever was the dominant water mass, we suggest that the weakening of the AMOC, reduction of deep-water ventilation and shoaling in the boundary between AABW and NADW as the main mechanism increasing storage of respired carbon during glacial intervals of the MPT. Nevertheless, we acknowledge in the discussion (L. 311-329) that alternative explanation tested for LGM exist, although the current proxy data available for Site U1314 during the MPT does not warrant confirmation of that scenario.

There is something unusual in the authors arguments about dysoxic bottom-pore water conditions. Why is the $\delta^{13}\text{C}$ record of cibs continuous as one would expect them not to be present at all considering their habitat behavior. Quantitative abundances of infaunal vs cibs would be very helpful in this matter.

Dysoxic conditions do not fully prevent the occurrence of oxic species such as those belonging to the genus *Cibicides*. It is possible to find *Cibicides* spp in low oxygen concentration environments (e.g. Burkett et al 2016). Moreover, we screened the whole sample for 1-2 well preserved individuals for isotopic analysis in the 150 micrometre fraction in contrast with the >125 micrometers fraction studied for faunal assemblages. Only one or two individuals are needed for the isotope analyses, whereby the whole size fraction >150 μm was screened to find individuals, in contrast to the faunal analyses that were done on a fraction of that size fraction. We, therefore, do not find *anything unusual in our data*. *The plot below shows how relative abundance of C. wuellerstorfi* (data which is in the supplementary material and archived in PANGAEA) in the assemblages reaches a maximum of 13%, and lowest values or absence are found in just a few samples, normally contemporarily with

more negative $\delta^{13}\text{C}_b$ (published data, see Hernandez-Almeida et al. 2013) values at Site U1314.

Also, like to see a planktonic $18\text{O}/13\text{C}$ record as my own experience shows a parallel trend in benthic and planktonic $\text{d}13\text{C}$, essentially meaning a clear vertical top-to-bottom (water column) connection. Thus that connection might be well through the food chain with changing surface productivity...etc. And indeed I checked the core's record and found that the authors have actually published (in 2015) that parallel trend into MIS24...! Thus I would like to see a complete 13C record of *N. pachyderma* and wonder why don't other parts with very low benthic $\text{d}13\text{C}$, such as MIS 30, also show the same response...?

As pointed out by the reviewer, the planktonic $\delta^{13}\text{C}$ data as well as the gradient between benthic-planktonic $\delta^{13}\text{C}$ were published in Hernández-Almeida et al (2015), which we paste below. In this figure, it can be observed that the trend in the gradient during MIS 30 and MIS 24 is similar. Indeed, during MIS 30 there is a similar response in the geochemical and faunal proxies at U1314, with values indicating lower BPWO, as well as a moderate decrease in O_2 as reconstructed from the benthic carbon isotope gradient at U13185. As indicated by the figure below the gradient decreases during periods with IRD deposition when a sea ice cover and/or a freshwater lens hampered a good ventilation of the surface waters near Site U1314. Although the Rockall Plateau region, potentially including the Iceland basin where site U1314 is located, has been indicated as a deep convection area during glacial times (e.g., during LGM: Duplessy et al., 1988; Sarnthein et al., 1994; Gherardi et al. 2005, or during MIS 12: Voelker et al. 2010), we cannot infer from our data that this was also the case during the MPT glacials. So, we cannot state that a top-to-bottom $\delta^{13}\text{C}$ signal transport occurred in the vicinity of Site U1314. A sea ice cover and freshwater lid would also be present in the upstream region of the Nordic Seas and in the Labrador Sea, i.e., those regions associated with deep convection today, so that the depleted surface water $\delta^{13}\text{C}$ signal could originate from further to the north or the west and be inherited by the NADW observed at Site U1314. Kleiven et al. (2003) found little similarity between planktonic and benthic $\delta^{13}\text{C}$ at Site 983 (which occupies almost the same position and depth as U1314), meaning that preformed changes in surface water $\delta^{13}\text{C}$ are not the dominant influence on Atlantic deep-circulation $\delta^{13}\text{C}$ changes. Since vertical top-to-bottom (water column) connection and changes in the surface ocean nutrient utilization does not fall directly on the main topic in our study, we have decided to leave this discussion out of the revised version.

Of course, to obtain an insight into the pore water and the bottom water, a comparison of $\delta^{13}\text{C}$ between infaunal species with epifaunal cibs seems necessary, perhaps supported by some $\delta^{13}\text{C}$ analyses of the bulk carbon in the sediment (see below).

Unfortunately, measurements of the $\Delta\delta^{13}\text{C}$ between paired foraminiferal species (including *C. wuellerstorfi* and *Globobulimina affinis* or *G. pacifica*) is not possible since the abundance of the infaunal species (*Globobulimina* spp.) needed for that proxy are extremely scarce (0.09% average abundance, and present only in 20 samples), in contrast to other sites in the North Atlantic. Using other infaunal species present at U1314 is problematic, since infaunal species do migrate into the sediment (Jorissen et al., 1999) in response to food supply and oxygen concentrations. Thus, any other $\delta^{13}\text{C}$ epifaunal-infaunal based proxy using an infaunal species besides *Globobulimina* sp. would have to be interpreted with caution because it could vary depending on the varying microhabitat of infaunal species within the sediment. Second, this calculation- which is not available- would not support further our arguments. It stands for reason that if the bottom water is dysoxic (as geochemistry supports), the pore water will be, at least dysoxic. Regarding the analyses of the bulk carbon in the sediments, we do not think that this measurement would be helpful to obtain any insight into the pore water and the bottom water, since this isotopic composition would be dominated by the carbonates of the most abundant fraction, which are upper ocean planktonic organisms at Site U1314, i.e. coccolithophores, followed by planktonic foraminifera (Channell et al 2006), with overprints of ice-rafted carbonates as documented by Hodell et al. (2008) who established the $\delta^{18}\text{O}$ of bulk carbonate in the mid-latitude North Atlantic's IRD belt as an indicator for IRD events.

I also wonder what is the impact of certain type of IRD (ie, carbon-rich sediment clasts) would have on the assumption of stagnated bottom waters? Their bulk TOC record might give insight, because the conclusion of the redox conditions could simply be, by and large, a post- sedimentary pore water process that happened much later and which had nothing to do with conditions at the sediment-water interface at the time of sediment deposition.

Unfortunately, we do not have data on the geochemical composition of the IRD grains, only the counts. However, we did not observe dolomite clasts within the coarse lithic fraction during the counting (made by the first author of this manuscript), being the dominant lithologies volcanic and quartz. Moreover, carbon-rich sediment clasts are not common at this latitude and time interval (Naafs et al. 2013). If the reviewer intends to suggest that burial and remineralization of TOC-rich sediments can drive the redox signal, this possibility can be excluded since the (inferred) low BPWO conditions occur during low TOC conditions (see supplementary information, L 133-154, and supplementary Fig. 3).

Between 780-960ka, it is intriguing to see (suppl. Fig 2) calcite and TOC(%) – latter being pretty low in general - to show such similar trends, and in comparison, the Mn/Al bulk mimics the calcite. Why is that?

We believe that the similarity between Mn/Al and calcite is causality. Higher calcite and TOC is recorded during warmer SST and higher primary productivity (Hernandez-Almeida et al. 2012;

Supplementary Fig. 3). During these more favourable climate conditions, there is also more intense deep-water convection and higher BPWO, leading to higher Mn/Al.

The paper is based on a lot of “maybes” and other guesses without any further in-depth groundtruthing. This should involve other interval(s) that had a similar glacial environment with very low $\delta^{13}\text{C}$ signature and an indicative benthic foraminiferal assemblage, but for which no dysoxic conditions are implied.

We propose – based on our data and other previously published research– a perfectly plausible scenario for explaining the low BPWO during cold, IRD-rich intervals in the subpolar North Atlantic, but we prefer being cautious in our hypothesis. Actually, during MIS 20, MIS 26, MIS 28 and MIS 30 there are similar low values in our geochemical (Mn/Al, P/Al) and benthic foraminifera data that could be inferred as dysoxic conditions. Indeed, downstream site U1385 records also O_2 reductions (see Thomas et al. 2022) during these intervals. We did not focus on those events, because the literature has not identified them as key time intervals in the climate reorganization of the MPT. Nevertheless, we now emphasize (e.g. at L. 231, 341, 398), that these these glacial maxima record also similar dysoxic conditions and the interpretation of changes in northern vs southern waters can be extended to those glacial intervals.

For instance, in nearby core 1313 from some greater water depth – this one btw should exhibit the same features during MIS24-22 – MIS 12 would be a suitable test as it also shows very low $\delta^{13}\text{C}$ values (published by co-author Voelker). I do appreciate suppl. Fig 1 but it only provides a crude glimpse into the issue for MIS12 or some other, longer intervals. But besides MIS12 there are many more suitable intervals, especially those from major glacial terminations (see records by Hodell et al. 2025) when the meltwater stratification hypothesis could be further tested.

We appreciate the suggestion, but extending our study to MIS 12 is beyond the scope of this research, because our set of proxies is limited to MIS 19-31, and the word limitation for manuscript in this journal.

Having said the above, I find the jump to the southern hemisphere premature and overly ambitious at this stage; the various proxy records selected is a bundle of very different type of “fruit” that might or might not have anything to do with each other or the northern record.

This opinion differs from rev#2, who says that the integration of different geochemical proxies from 1094 and U1314 is commendable, provides a well-structured discussions it enhances the robustness of the interpretations regarding bottom water oxygenation (see point 7 of rev#2). Moreover, we find that our study is novel in the sense that it is the first time that similar proxies have been compared at both polar regions. We agree with the reviewer in that north and south hemisphere are of different nature, but we provide arguments to demonstrate that proxy records from U1314 and 1094 (i.e. planktonic foraminifera Mn/Ca) are indicative of BPWO at both locations (see Supplementary Fig. 1), and that sedimentary conditions are not conditioning the interpretation of proxies at both locations (See section “sedimentary environments at Sites U1314 and 1090” in the Supplementary information).

At present I see the investigation and thoughts provided in the manuscript as a solid case study to start off with. However, it would require more in-depth and broader testing of the proxies applied before any globally relevant implications on the ocean-climate system, and here especially the potential involvement of pCO_2 during MIS24-22 and the further MPT development, are warranted. The latter one in particular I find really hard to accept at present.

We agree with the reviewer that having more data, particularly high-resolution pCO_2 records during MIS 24-22 from ice-cores, would be ideal to test any potential involvement (and quantitative) of polar regions in global climate. Lacking pCO_2 records/high resolution reconstructions across the MPT prevent us to explore further the pCO_2 role in the MPT. Instead, we focus on the role of the high-latitude ocean-ice-sheet interaction on the BPWO in the ocean and its potential role in global climate. We hope that our work helps further studies to generate new proxies that help to ground this hypothesis.

References

- Bates, Stephanie L., Mark Siddall, and Claire Waelbroeck. "Hydrographic variations in deep ocean temperature over the mid-Pleistocene transition." *Quaternary science reviews* 88 (2014): 147-158.
- Becquey, Sabine, and Rainer Gersonde. "Past hydrographic and climatic changes in the Subantarctic Zone of the South Atlantic—The Pleistocene record from ODP Site 1090." *Palaeogeography, Palaeoclimatology, Palaeoecology* 182.3-4 (2002): 221-239.
- Bianchi, G. and I. McCave, Hydrography and sedimentation under the deep western boundary current on Björn and Gardar Drifts, Iceland Basin. *Marine Geology*, 2000. 165(1-4): p. 137-169.
- Burkett, Ashley M., et al. "Colonization of over a thousand *Cibicides wuellerstorfi* (foraminifera: Schwager, 1866) on artificial substrates in seep and adjacent off-seep locations in dysoxic, deep-sea environments." *Deep Sea Research Part I: Oceanographic Research Papers* 117 (2016): 39-50.
- Channell, J. E. T., and J. S. Stoner. "Plio-Pleistocene magnetic polarity stratigraphies and diagenetic magnetite dissolution at ODP Leg 177 Sites (1089, 1091, 1093 and 1094)." *Marine Micropaleontology* 45.3-4 (2002): 269-290.
- Channell, J. E. T., Kanamatsu, T., Sato, T., Stein, R., Alvarez Zarikian, C. A., Malone, M. J. & Expedition 303/306 Scientists 2006: Expedition 306 summary. In J. E. T. Channell, T. Kanamatsu, T. Sato, R. Stein, C. A. Alvarez Zarikian, M. J. Malone & Expedition 303/306 Scientists (eds.): *Proceedings Integrated Ocean Drilling Program, Expedition 303/306*, 1–29. Integrated Ocean Drilling Program, College Station, TX.
- Curry, W.B., Duplessy, J.C., Labeyrie, L.D., Shackleton, N.J., 1988. Changes in the distribution of $\delta^{13}\text{C}$ of deep water ΣCO_2 between the last glaciation and the Holocene. *Paleoceanography* 3, 317-341, doi: <https://doi.org/10.1029/PA003i003p00317>.
- Curry, W.B., Oppo, D.W., 2005. Glacial water mass geometry and the distribution of $\delta^{13}\text{C}$ of ΣCO_2 in the western Atlantic Ocean. *Paleoceanography* 20, PA1017, doi: 10.1029/2004PA001021.
- Diz, Paula, et al. "Paleoceanography of the southern Agulhas Plateau during the last 150 ka: Inferences from benthic foraminiferal assemblages and multispecies epifaunal carbon isotopes." *Paleoceanography* 22.4 (2007).
- Duplessy, J.C., Shackleton, N.J., Fairbanks, R.G., Labeyrie, L.D., Oppo, D., Kallel, N., 1988. Deepwater source variations during the last climatic cycle and their impact on the global deepwater circulation. *Paleoceanography* 3, 343-360,
- Gherardi, J.M., Labeyrie, L., McManus, J.F., Francois, R., Skinner, L.C., Cortijo, E., 2005. Evidence from the Northeastern Atlantic basin for variability in the rate of the meridional overturning circulation through the last deglaciation. *Earth and Planetary Science Letters* 240, 710-723
- Gherardi, J., Labeyrie, L., Nave, S., Francois, R., McManus, J.F., Cortijo, E., 2009. Glacial - interglacial circulation changes inferred from 231Pa/230Th sedimentary record in the North Atlantic region. *Paleoceanography* 24, doi: 10.1029/2008PA001696.
- Eberwein, Astrid, and Andreas Mackensen. "Regional primary productivity differences off Morocco (NW-Africa) recorded by modern benthic foraminifera and their stable carbon isotopic composition." *Deep Sea Research Part I: Oceanographic Research Papers* 53.8 (2006): 1379-1405.
- Ferrari, Raffaele, et al. "Antarctic sea ice control on ocean circulation in present and glacial climates." *Proceedings of the National Academy of Sciences* 111.24 (2014): 8753-8758.
- Filippelli, Gabriel M., and Margaret Lois Delaney. "Phosphorus geochemistry of equatorial Pacific sediments." *Geochimica et Cosmochimica Acta* 60.9 (1996): 1479-1495.
- Gherardi, J-M., et al. "Glacial-interglacial circulation changes inferred from 231Pa/230Th sedimentary record in the North Atlantic region." *Paleoceanography* 24.2 (2009).

- Gottschalk, Julia, et al. "Carbon isotope offsets between benthic foraminifer species of the genus *Cibicides* (*Cibicidoides*) in the glacial sub-Antarctic Atlantic." *Paleoceanography* 31.12 (2016): 1583-1602.
- Hernández-Almeida, I. et al. "Palaeoceanographic changes in the North Atlantic during the Mid-Pleistocene Transition (MIS 31–19) as inferred from planktonic foraminiferal and calcium carbonate records." *Boreas* 42.1 (2012): 140-159.
- Hernández-Almeida, I., et al. "A high resolution opal and radiolarian record from the subpolar North Atlantic during the Mid-Pleistocene Transition (1069–779 ka): Palaeoceanographic implications." *Palaeogeography, Palaeoclimatology, Palaeoecology* 391 (2013): 49-70.
- Hines, Sophia KV, et al. "Revisiting the mid-Pleistocene transition ocean circulation crisis." *Science* 386.6722 (2024): 681-686.
- Hodell, David A., et al. "Onset of "Hudson Strait" Heinrich events in the eastern North Atlantic at the end of the middle Pleistocene transition (~ 640 ka)." *Paleoceanography* 23.4 (2008).
- Howe, Jacob NW, et al. "North Atlantic deep water production during the Last Glacial Maximum." *Nature communications* 7.1 (2016): 11765.
- Jorissen F. J. (1999) Benthic foraminiferal microhabitats below the sediment-water interface. In *Modern Foraminifera* (ed. B. K. Sen Gupta). Kluwer Academic Publishers. pp. 161–179.
- Kanfoush, Sharon L., et al. "Comparison of ice-rafted debris and physical properties in ODP Site 1094 (South Atlantic) with the Vostok ice core over the last four climatic cycles." *Palaeogeography, Palaeoclimatology, Palaeoecology* 182.3-4 (2002): 329-349.
- Kim, J., et al., North Atlantic deep water during Pleistocene interglacials and glacials. *Quaternary Science Reviews*, 2021. 269: p. 107146.
- Kleiven, Helga Flesche, et al. "Atlantic Ocean thermohaline circulation changes on orbital to suborbital timescales during the mid-Pleistocene." *Paleoceanography* 18.1 (2003).
- Lynch-Stieglitz, Jean, et al. "Atlantic meridional overturning circulation during the Last Glacial Maximum." *science* 316.5821 (2007): 66-69.
- Mackensen, Andreas, et al. "The $\delta^{13}\text{C}$ in benthic foraminiferal tests of *Fontbotia wuellerstorfi* (Schwager) relative to the $\delta^{13}\text{C}$ of dissolved inorganic carbon in southern ocean deep water: implications for glacial ocean circulation models." *Paleoceanography* 8.5 (1993): 587-610.
- Mangini, A., Anton Eisenhauer, and P. Walter. "Response of manganese in the ocean to the climatic cycles in the Quaternary." *Paleoceanography* 5.5 (1990): 811-821.
- McManus, James, et al. "Authigenic uranium: relationship to oxygen penetration depth and organic carbon rain." *Geochimica et Cosmochimica Acta* 69.1 (2005): 95-108.
- Moffitt, S. E., Moffitt, R. A., Sauthoff, W., Davis, C. V., Hewett, K., & Hill, T. M. (2015). Paleoclimatographic insights on recent oxygen minimum zone expansion: Lessons for modern oceanography. *PloS One*, 10(1), e0115246. <https://doi.org/10.1371/journal.pone.0115246>
- Muglia, Juan, and Andreas Schmittner. "Carbon isotope constraints on glacial Atlantic meridional overturning: Strength vs depth." *Quaternary Science Reviews* 257 (2021): 106844.
- Naafs, B. D. A., Jens Hefter, and Ruediger Stein. "Millennial-scale ice rafting events and Hudson Strait Heinrich (-like) Events during the late Pliocene and Pleistocene: a review." *Quaternary Science Reviews* 80 (2013): 1-28.
- Raymo, Maureen E., et al. "Stability of North Atlantic water masses in face of pronounced climate variability during the Pleistocene." *Paleoceanography* 19.2 (2004).

- Repschläger, Janne, et al. "Active North Atlantic deepwater formation during Heinrich Stadial 1." *Quaternary Science Reviews* 270 (2021): 107145.
- Ruttenberg, Kathleen C., and Robert A. Berner. "Authigenic apatite formation and burial in sediments from non-upwelling, continental margin environments." *Geochimica et cosmochimica acta* 57.5 (1993): 991-1007.
- Sarnthein, M., Winn, K., Jung, S., Duplessy, J., Labeyrie, L., Erlenkeuser, H., Ganssen, G., 1994. Changes in east Atlantic deepwater circulation over the last 30,000 years: Eight time slices reconstructions. *Paleoceanography* 9, 209-267.
- Tachikawa, K. et al. (2021): Eastern Atlantic deep-water circulation and carbon storage inferred from neodymium and carbon isotopic compositions over the past 1.1 million years. *Quaternary Science Reviews*, 252, 106752
- Tribovillard, Nicolas, et al. Trace metals as paleoredox and paleoproductivity proxies: an update. *Chemical Geology* 232.1-2 (2006): 12-32.
- Thunell, R.C., Poli, M.S., Rio, D., 2002. Changes in deep and intermediate water properties in the western North Atlantic during marine isotope stages 11-12: evidence from ODP Leg 172. *Marine Geology* 189, 63-77,
- Van der Zwaan, G. J., et al. "Benthic foraminifers: proxies or problems?: a review of paleocological concepts." *Earth-Science Reviews* 46.1-4 (1999): 213-236.
- Voelker, Antje HL, et al. "Variations in mid-latitude North Atlantic surface water properties during the mid-Brunhes (MIS 9–14) and their implications for the thermohaline circulation." *Climate of the Past* 6.4 (2010): 531-552.
- Wang, Yi, et al. "Global oceanic oxygenation controlled by the Southern Ocean through the last deglaciation." *Science Advances* 10.3 (2024): eadk2506.
- Wang, P., Tian, J., Cheng, X., Liu, C., Xu, J., 2003. Carbon reservoir changes preceded major ice-sheet expansion at the mid-Brunhes event. *Geology* 31, 239-242
- Wang, P., Tian, J., Lourens, L.J., 2010. Obscuring of long eccentricity cyclicity in Pleistocene oceanic carbon isotope records. *Earth and Planetary Science Letters* 290, 319-330
- Yu, Jimin, et al. "Preferential dissolution of benthic foraminiferal calcite during laboratory reductive cleaning." *Geochemistry, Geophysics, Geosystems* 8.6 (2007).
- Yehudai, M., et al., Evidence for a Northern Hemispheric trigger of the 100,000-y glacial cyclicity. *Proceedings of the National Academy of Sciences*, 2021. 118(46): p. e2020260118

REVIEWER COMMENTS

Reviewer #1 (Remarks to the Author):

This is the second time I have seen this manuscript. Overall, I think that it is much improved but could still use some editing. The authors present an interesting interpretation of their data in the context of other published results that does help to explain transitions during the MPT while also pointing to areas where new data is needed. I am satisfied with how the authors have dealt with reviewer comments. Specific suggestions for improvement are listed here:

We thank the reviewer for the constructive assessment and additional suggestions for improvement, particularly the editing of some of the sentences which add clarity to the manuscript. We will carefully address each point to further strengthen the manuscript.

Beginning line 40 - Last sentence of the first paragraph needs to be edited. Suggestion: The deep ocean, Earth's largest carbon reservoir that readily exchanges with the atmosphere, has been proposed to play a key role in carbon sequestration and could mitigate the climate change risks of global warming.

Thanks for the suggestion, replaced accordingly.

Line 44 – suggest reorganization: The leading hypothesis to explain the MPT suggests that the initiation of 100-kyr glacial cycles was driven by increased carbon storage in the deep ocean.

Thanks for the suggestion, replaced accordingly.

Line 45 – The Southern Ocean has emerged as the leading region to explain the climate feedbacks during the MPT.

Thanks for the suggestion, replaced accordingly.

Line 49 – remove “the” in before Antarctic

Removed.

Line 50 – remove “ on the idea”

Deleted.

Line 52 – change to “Enhanced glacial marine carbon sequestration...”

Thanks for the suggestion, replaced accordingly.

Line 53 – change “deoxygenation” to “reduced oxygen”

Changed accordingly.

Sentence beginning line 56 – add “which” before imply or some other rewording is necessary

We guess the reviewer refers to L. 58. Word added.

Line 62 – “should” or “would”? then “would” becomes “could”?

“Would” replaced by “could”.

Line 82 – “New and previously published proxy records”?

Changed accordingly. We have added BPWO proxy records as per requested by reviewer#4.
Line 87 – missing “)”

Closing bracket added.
Line 93 – “Shifting”

Changed accordingly.
Line 96 – change “are argued to reflect” to “may reflect”

Changed accordingly.
Line 106 – change “those” to “these”

Replaced accordingly.
Line 111 – and “a” before Mn-rich

“a” added.
Sentence beginning 113 -break into multiple sentences. “...increases at the glacial/interglacial transitions. At U1314, these increases also occur during interglacial periods. Increased Mn/Ca is indicative of more oxide conditions, ...”

Sentence replaced into multiple sentences.
Line 132 – “at” should be “in all oceans”

Replaced.
Sentence beginning 131 should be split – New sentence after [53], “These observations eliminate temperature-driven oxygen...”

New sentence added and reworded.
Line 135 remove “,” before “or”

Comma deleted.
Line 177 – either “increases in ventilation” or “increased ventilation”

We opted for “increased ventilation”
Sentence beginning line 213 – The timing of the decrease in Mn/Al. and P/Al ratios and reactive phases displayed by the U1314 records is also coeval with the largest decreases in BPWO...”

Replaced accordingly.
Line 220 – “used”? should this be “observed” or possibly just removed and read as “compared to U1385”

Replaced accordingly.
Line 224 – “which could be related...”

“Be” has been added.
Line 227 – “suggest substantial shifts” or suggest a substantial shift”

Sentence changed: “a substantial shift”.

Line 228 – remove “anyways”

Removed.

Line 233 – “reductions in deep-water...”

“In” added.

Line 249 – a word is missing before geochemical

We have deleted “geochemical”.

Line 260 “will” should be “would”

Replaced.

Sentence beginning line 323 – should be split into two sentences

We started a new sentence after “MIS 26”.

Line 333 – “critical data” or “critical evidence” not “critical data evidence”

We have left “critical evidence”.

Line 362 – why not give the actual number of samples?

Actually there was a mistake, the total number of samples has been added (298).

Line 364 – use approximately here instead of around

Changed.

Line 368 – I think the samples were diluted to 50 mL rather than diluted with 50 mL of Milli-Q

Replaced.

Line 370 – “in the Department...”

Replaced.

In the methods/elemental geochemistry section, it would be useful to include data about precision and accuracy either in the main text or the supplemental material

The external precision is detailed in L. 524-525.

Line 377 – I’m not sure that is a valid assumption. Other studies have shown that there are significant differences in provenance on glacial/interglacial time scales.

We added a reference to support this statement, Calvert et al (2007), and modified the assumption by saying that Al normalization is recommended due to its conservative behaviour in marine sediments.

Line 406 – “being” should be “and the reproducibility of all phases was on average...”

Replaced.

Line 420 – “weighed” not weighted

Replaced.

I like the arrows on Fig. 3 – can there be arrows added to Fig. 2 to help explain what we are supposed to see?

Arrows showing the direction of the proxies have been added also to fig. 2.

The figure caption for supplementary figure 3 does not include the Site.

This information has been added.

Reviewer #2 (Remarks to the Author):

I thank the authors' thoughtful and thorough responses to my previous comments. The manuscript has been thoroughly revised and substantially improved, especially in clarifying proxy limitations, sharpening the discussion, and better integrating supplementary materials into the main text. I have a few remaining comments that I think could further strengthen the manuscript before it's ready for publication:

We thank the reviewer for their positive assessment of the revisions and for recognizing the improvements made to the manuscript. We appreciate the reviewer's careful re-evaluation and their acknowledgement of our efforts to clarify proxy limitations, refinement of the discussion and integration of supplementary materials in the main text.

Below, we provide detailed responses to each point raised in the new round of reviews and outline the corresponding revisions made in the text.

Because the authors compare BPWO changes in both polar regions during the MPT and point to similar patterns, it raises a question: which region do you think played a bigger role in driving the MPT? Was the North Atlantic or Southern Ocean?

This is an interesting question, and we try to give our opinion in L. 460-470, by saying that no single region had a bigger role, but both polar regions and associated climate-ocean feedbacks work together. Until now, studies were focusing on the role of the Southern Ocean driving the MPT, but our study adds new evidence to complete the puzzle. We recognize, based on the current evidence, that the Southern Ocean might have a leading role, but in this study, we demonstrate that changes in the subpolar North Atlantic were essential to allow a greater oceanic carbon storage during the MPT.

The authors have expanded the discussion of MIS 23 $\delta^{13}\text{C}$ values and multi-proxy patterns. However, the link to broader MPT climate dynamics feels a bit underplayed. Since MIS 23 seems to buck the trend of reduced BPWO, could it be signaling a partial recovery in deep-water ventilation? Might this interglacial represent some kind of tipping point or a transitional interval within the MPT progression? A bit more speculation or contextualization here could be insightful.

We thank the reviewer for asking for additional speculation regarding potential tipping points during MIS 23. MIS 23 has been identified as a failed interglacial by Elderfield et al. (2012),

with critical step in global ice-volume variation and lower summer insolation at 65°N and S which led to the suppression of melting in MIS 23. So yes, we believe that MIS 23 and its configuration allowed the first 100-ky glacial cycle comprising MIS 22 to 24 which represents a tipping point in AMOC geometry and therefore a tipping point in BPOW. However, because this opinion contrasts with that of reviewer 4, who disagrees with the speculative tone of the manuscript, we prefer not adding additional speculation in the manuscript.

Overall, I support publication after these remaining points are addressed. This work provides a meaningful contribution to our understanding of bottom water oxygen variability during the MPT.

Reviewer #4 (Remarks to the Author):

Review of MS submission of Hernández-Almeida et al. entitled “Glacial dysoxia in the deep subpolar North Atlantic during the Mid-Pleistocene Transition”

Hernández-Almeida et al. present an impressive amount of data from IODP Site U1314 from the North Atlantic addressing climate- and carbon cycle dynamics during the MPT. This is a very important long-term climate transition that manifested itself without any notable changes in the external (i.e., orbital) forcing, pointing at the importance of internal climate feedback mechanisms. The central motivation of the study of Hernández-Almeida et al. is to better characterize these internal climate feedback mechanisms through the reconstruction of redox-sensitive proxies in the marine sediment record of IODP U1314, particularly illuminating possible changes in North Atlantic carbon storage variations and how they are linked with northern-hemisphere ice sheet dynamics and Atlantic overturning dynamics. Here I provide my evaluation of the study, joining the group of reviewers after the first revision round. I cannot comment in detail on the comments of the other reviewers, but overall the paper is too speculative in places and the main conclusions are not sufficiently backed up with proxy evidence.

Thanks to the reviewer for the comments, which definitively motivated us to improve the manuscript. We tried to find a balance between what our multi-proxy dataset indicates and what processes for which we do not have direct evidence could cause these changes. We would like to point out that some of the comments by reviewer #4 are opposite to reviewer’s #2 opinion, who in some places recommends “A bit more speculation or contextualization here could be insightful”, particularly regarding the role of the Southern vs Northern Hemisphere. We justify those arguments which are considered sometimes “speculative” by reviewer 4 by more clearly referencing the studies and data that support the interpretation of our study. In addition, we implemented almost all minor suggestions by reviewer 4.

The study is an impressive amount of work, and I would like to congratulate the authors for this study. I do, however, think that the way the main hypotheses and argumentation is laid out and the figures are presented is not fully convincing. Reviewer #3 has many concerns about the mechanistic concepts that the authors suggested, and I must admit that I share these concerns still.

Thank you for praising our multi-proxy record. Regarding the reviewer #3 concerns, we have answered those issues in our previous rebuttal letter and explained the mechanistic concepts behind our interpretation. Please, refer to the review of the “first round” where we specifically replied to reviewer #3. This review is to specifically reply to #review 4 comments. Here, we will try to clarify the mechanistic concepts once again and improve the manuscript whenever it is necessary.

Major comments

One of my biggest concerns of the study is the fact that the time interval of study is limited to 780 – 1180 kyr. This covers the MPT, while many might argue that an important part of the MPT is missing (e.g. MIS 16). The authors argue that carbon cycle changes in the North Atlantic contributed “to the pCO₂ decrease during the MPT” (abstract). However, the authors provide no indication that these observations of bottom water oxygen changes in the North Atlantic persist post 780 kyr before present, which would explain the long-term drop in atmospheric CO₂ during post-MPT glacials (Hönisch et al., 2009; Chalk et al., 2017). In that sense, I find this conclusion of the study too bold, given the absence of data post MPT.

As commented before in our reply to reviewer #3, extending our study to other glacial intervals is beyond the scope of this research. Besides, the MPT, as per defined by several significant studies on the MPT, goes from 1,250-1200 and was completed between 800-700 ka (Herbert et al 2023; Clark et al 2006; Chalk et al 2017; Hines et al 2024; Hönisch et al 2009). The most recent review (Herbert et al. 2023) does not include MIS 16 within the MPT. Therefore, even if this study does not encompass the whole proposed range for the MPT, the time interval studied in this study is considered crucial for the MPT because it includes the thermohaline circulation “crisis” at 900 ka (Marine Isotope Stages 24 to 22) (Pena et and Goldstein 2014). In conclusion, the time interval of this study is considered key for understanding the process taking place at the core of the MPT. Aiming to sustain more the value of the time frame of our research we show below a comparison of the XRF Mn/Al data from IODP Site U1602 (blue line in the figure below; Parnell-Turner et al. 2025, located at 61°11.7138'N, 38°10.8186'W, 2710 mbsf, age model based on bio and magnetostratigraphy), with the bulk Mn/Al from IODP Site U1314 (orange line), both proxies of bottom water oxygenation. The correspondence between high (interglacial) and low (glacial) Mn/Al values for the studied interval, particularly the very low values between 870-930 kyr (MIS 24-22) is remarkable, meaning that similar changes in BPWO are found across the subpolar North Atlantic. Lower XRF Mn/Al values at Site U1602 are also recorded at MIS 16 and 18, although not as low as during MIS 24-22, meaning that the most important reduction in BPWO occurred during the former interval. If deemed necessary by the reviewers, this figure will be added to the final version of the manuscript, in the supplementary material, but for the moment, we do not consider it is necessary to expand the supplementary material further.

The authors set out to test the “leading hypothesis” of the MPT being driven by deep ocean carbon changes. Please provide references for studies establishing this hypothesis (l. 44-45). I do not find the results of the study novel enough given the uncertainties regarding the mechanisms put forward (no proxy evidence for surface ocean salinity decline or changes in AMOC strength). The findings of the study in terms of bottom water oxygen variations are not very different from the findings of Lear et al. (2016) [ref. 27] and Thomas et al. (2022) [ref. 28]. Of course, it is always beneficial to get a basin-wide view.

References for studies establishing the deep ocean carbon changes as the main hypothesis to explain the MPT have been added. We disagree with the idea that our study is not novel enough, either due to absence of enough proxy evidence or compared to findings by the papers cited by the reviewer.

Firstly, we do not claim changes in the AMOC strength based on our multi-proxy dataset, but changes in the AMOC depth (e.g. L. 254). To distinguish between AMOC depth and strength, it would be necessary a three-dimensional picture provided by several sites and water depth (Muglia and Schmittner 2021).

However, and once again, we think the data provided in this study together with previously published proxies allow us to constrain the role of changes in the AMOC to explain the BPWO variability. Thus, to support even further our discussion, we now compare in supplementary figure 6A the XRF K/Ti ratio from Site U1314 (Gruetzner and Higgins 2010) with some of our BPWO proxies and provide additional arguments in L. 248-254. The changes in XRF K/Ti are indicative of changes in the sediment provenance controlled by the variable flow of the ISOW (Gruetzner and Higgins, 2010). Low K/Ti values would be indicative of a vigorous ISOW flowing south over the Iceland-Faroe Ridge delivered high amounts of basaltic material to the Gardar Drift, whereas high K/Ti values during glacial and millennial-scale cold events indicates a different source, likely more from the southeast, meaning that the core of the ISOW shoaled and sedimentation at the depth of Site U1314 (as well as other sites in the Gardar Drift) became influenced by enhanced southern-sourced waters flow (Gruetzner and Higgins, 2010; Stevenard et al. 2024; Sinnesael et al. 2025). The similarity between our bulk sediment Mn/Al and the XRF K/Ti indicates that our BPWO proxies are controlled by dynamical changes in the deep-water masses bathing Site U1314 (i.e. ISOW). In the same way, there is a good agreement between low benthic $\delta^{13}\text{C}$ values and high XRF K/Ti ratios, particularly during MIS 30, 26, 22 and 20.

New Supplementary Fig. 6:

Secondly, regarding the novelty of our study overall, reviewer's #4 opinion contrast with that of reviewer 1, who says that our study "does help to explain transitions during the MPT while also pointing to areas where new data is needed".

The general understanding is that Southern Ocean processes dominated the climate feedbacks during the MPT (see L. 57-62 in the revised manuscript and references therein). However here, our results indicate that cryosphere-ocean-climate dynamics in the subpolar North Atlantic played a critical role in regulating deep-ocean circulation and thereby facilitating carbon

sequestration across the entire Atlantic basin. This is the first time that a multiproxy reconstruction allows for interpreting a key role of the subpolar North Atlantic in driving carbon cycle changes during the MPT.

The reviewer mentions that our results are not novel and reference a few previous papers. We disagree with the reviewer and provide further explanation below:

Lear et al. (2016) and Thomas et al. (2022) studied cores located between 37.5 °N and 41°N (not in the subpolar Atlantic), although both somehow inferred that sea ice and surface stratification played a role in the reduced deep-water oxygen. Significantly, none of them provided proxy data to support the hypothesis:

Lear et al. (2016) inferred increased carbon sequestration in the deep-as a result of an increase in glacial sea ice extent in the Southern Ocean although no proxy evidence or record from the Southern Ocean showing concomitant decreases in BPWO in deep-water source areas in the Southern Ocean are shown in that study. These authors suggest that the “increased storage of carbon in the deep North Atlantic as southern-sourced waters displaced northern-sourced waters during glacial intervals across the MPT”. We build-up on this idea and provide the proxy evidence in the North Atlantic that shows that the deep-water mass geometry change was not a mere “passive” displacement of waters, but other processes involving cryosphere-ocean-climate changes near the source areas of deep-water formation in the Northern Hemisphere were involved.

Thomas et al. (2022) state that “the deep-water oxygen depletions may have been driven by increased freshwater inputs and ice rafting to the source areas of NADW formation”, but also opens the possibility that “the changes in deep-water [O₂] at Site U1385 could equally be driven by surface processes in the Southern Ocean such as changes in productivity”. Therefore, while Site U1385 provides a valuable quantitative O₂ reconstruction at U1385 this study is not conclusive regarding which process is driving the reduction in BPWO, a southern or northern origin.

In conclusion, we demonstrate in our study the interplay between changes in bottom water oxygen, changes in surface oceanography (southward shift of the arctic front) and the supply of IRD to the study site. This involves a mechanistic link between meltwater supply through decaying northern-hemisphere ice sheets, AMOC geometry and bottom water changes in the North Atlantic. We consider that our study helps to provide not only a basin-wide view, but also a mechanistic process at high-latitudes in the north Atlantic to explain changes in oxygenation in the deep ocean, and establish a relationship with processes (BPWO and sea ice expansion) in the SO. We think that we have tackled the remaining questions regarding the importance of polar regions in such BPWO reduction during the MPT, not fully addressed in the other studies mentioned by reviewer #4 which are in mid-latitudes in the North Atlantic.

I believe the core finding of the study is the agreement between changes in bottom water oxygen and the supply of IRD to the study site, implying a mechanistic link between meltwater supply through decaying northern-hemisphere ice sheets, AMOC geometry and bottom water changes in the North Atlantic. I find these assertions very interesting but speculative at the same time, in particular because direct evidence of changes in surface and/or bottom water salinities at the study site (e.g. seawater d18O) or convincing subpolar Atlantic sea ice variations or changes in

the locus of NADW formation is not provided. Possibly, Atlantic overturning strength did not decline during post-MPT glacials (as more and more LGM evidence suggests) but it shallowed and U1314 is influenced by southern-sourced water masses. Generally, I think some of the assertions that are the backbone of the study's outcome are not sufficiently backed up with proxy evidence.

As mentioned before, studies in the mid-latitude North Atlantic invoking changes in surface stratification as a mechanism driving reduction in deep-water oxygen. These studies invoke sea-ice and/or freshwater inputs and ice rafting to the source areas of NADW formation. Our study uses proxy data which can be used to infer these changes in surface ocean conditions that would lead to a reduction in the deep-water formation and consequent decrease in BPWO. In this regard, we disagree on the reviewer idea of “not backed up with proxy evidence”.

Three proxies are used to infer this:

- Record of sea ice variations at Site U1314 are supported by the record of the radiolaria species *Cycladophora davisiana*. According to Morley and Hays (1983) and (Bjorklund and Ciecieski 1994) the abundance of this species increases with the development of a strong low-salinity surface layer associated with seasonal sea ice melting, as observed today in the Okhotsk Sea (Okazaki et al. 2003)
- Melting related to IRD delivery. The occurrence of peaks in IRD/g suggest northern hemisphere ice-sheet retreat and meltwater supply to the subpolar North Atlantic at those times. The melting and destruction of icebergs releases freshwater to the oceans (Andrews 2000). Enhanced iceberg survivability due to older SSTs as the main driver of the IRD peak can be excluded, because temperatures were low at the time of the IRD delivery. The assumption that fresh water released as the result of the melting of icebergs disrupted deep-water formation lies in the foundation of our understanding of past global climate changes (Broecker 1994). At Site 983, which is located very close to U1314, in the Gardar Drift, as well as at sites 982 and 980, all located in the Northeast Atlantic, IRD/g is used as an indicator of freshwater derived from melting iceberg (Barker et al. 2015; Venz et al 1999; McManus et al. 1999). Therefore, in absence of specific proxies to reconstruct surface ocean salinity, we consider that the assumption that there was meltwater deliver at times of IRD deposition at Site U1314 is not speculative. In addition, we have cited at L. 203 the observation McClymont et al. (2008) who inferred the presence of cooler and fresher at the NE Atlantic during glacial intervals during the MPT at nearby ODP Site 983 based on the higher relative abundance of the C37:4 alkenone, which agrees with our inference based on IRD/g.
- Reviewer 4 believes that *N. pachyderma* is only influenced by SST; however, the recent distribution of *N. pachyderma* in the North Atlantic is indeed linked to the Greenland Current, which has a distinctly lower salinity and low surface $\delta^{18}\text{O}$ due to its source in the Arctic that differs greatly from the high salinity of the North Atlantic Current originating from the tropics. There is obviously also a temperature difference between the two water masses, but it is unclear which is more influential in the distribution of this species. In any case, there is no doubt about the strong relationship between IRD, low salinity and low $\delta^{18}\text{O}$ in surface waters in the North Atlantic. We agree that *N. pachyderma* is no direct sea-ice indicator, although it is the only “true-polar” species since 1 Myr ago (Weitkamp et al 2025) and it is capable of living in the ice-covered

Arctic Ocean (Carstens and Wefer, 1992). To clarify this and to avoid misinterpretation, we have revised L. 210, and specified that this species is an indicator of the position of the Arctic Front (following Wright and Flower, 2002), and polar waters. The important factor here is the presence of low salinity coming from meltwater that interrupts deep water formation. It is clear today there is a straight correlation between the low salinity, low surface $\delta^{18}\text{O}$, abundance of *N. pachyderma* and cold water, all of this associated to the Greenland current flowing from the Arctic (Kretschmer et al. 2016).

Much of the assertion of AMOC and/or NADW changes hinges on the interpretation of the benthic $\delta^{13}\text{C}$ record, but this proxy does not fingerprint regions of change very well and is influenced by multiple factors (including air-sea gas exchange).

We understand that the reviewer missed the Supplementary section (“Benthic carbon isotope versus redox records at U1314”), where we discussed the limitations of the benthic $\delta^{13}\text{C}$. Please, we kindly refer him/her to that section.

Generally, phases of decreased bottom-water $\delta^{13}\text{C}$ in the deep North Atlantic correlate with weakened AMOC phases, and vice versa. This inference is in agreement with complementary water-mass tracers (eNd records across the Atlantic showed in fig. 3) and basin-wide data (Extended data). Therefore, it does not seem premature to argue that Site U1314 was affected by changes in NADW. Besides we kindly refer the reviewer to our reply to his/her comment about the “AMOC strength” and the use of XRF K/Ti as additional ISOW proxy. Therefore, putting all the information together, it is reasonable to discuss AMOC changes at Site U1314.

Line 145-148: I find particularly speculative. IRD abundances are interpreted to indicate fresher surface waters, which enhanced surface buoyancy and limit the potential for open ocean deep convection. I presume in the North Atlantic. The only evidence is IRD and benthic $\delta^{13}\text{C}$. Other likely explanations for the benthic $\delta^{13}\text{C}$ signal are disregarded here, such as end-member $\delta^{13}\text{C}$ changes of deep water masses (thermodynamics during air-sea gas exchange) and/or the increased contribution of southern sourced water masses. Linking sea ice variations to the abundance of *N. pachyderma* is too bold in my view (as this has been traditionally considered as a temperature proxy). *N. pachyderma* can live in sea ice, but it is certainly not “ice related”. Any suggestion on changes in deep ocean ventilation as a result of surface ocean meltwater supply from BPWO is too speculative and should be avoided.

We kindly refer reviewer #4 to our reply about “Melting related to IRD delivery”, and the interpretation of *Neogloboquadrina. pachyderma* included above, as well as the alternative explanations for the changes in benthic $\delta^{13}\text{C}$ and additional NADW proxies included in our reply to his/her comment above.

Besides, previous publications have made similar links between IRD events and O_2 depletions, with an even more limited set of proxies (which in any case did not include seawater $\delta^{18}\text{O}$). The potential of IRD and associated meltwater release to shut down deep-convection in high latitudes was already suggested by Thomas et al. (2022), who stated that “the close association between [O_2] depletions and IRD events suggests that increased stratification and sea ice cover in NADW source regions reduced the oxygen supply to much of the deep North Atlantic”. Therefore, we do

not consider our statements speculative in any case or premature saying that there is a connection between meltwater supply from icebergs, changes in deep-ocean ventilation and reduction in BPWO at Site U1314. In addition, combining IRD and benthic $\delta^{13}\text{C}$ has been proven a robust approach to study surface water (melting input)–deep water structure, as shown by Starr et al. (2021).

The authors acknowledge different ways of interpreting the U1314 benthic $\delta^{13}\text{C}$ record (both as a signal coming from southern-sourced bottom waters and a changes in the geochemical signature of northern-sourced waters masses). They say “we are unable to clearly identify the water mass or process responsible for the decrease in BPWO”, which is bold and honest. Any suggestion that follows between lines 202-205 (and in fact in the Abstract) and in lines 230-234 remains thus a speculation in my view. The rest of the paper, despite this limitation, holds onto the interpretation of “weakening and shallowing of the AMOC upper cell” despite the limitations mentioned above. This is a little bit of a conundrum. Assertions such as in the response to Reviewer #1 (comment 1) that the authors show “that coupling between ocean and cryosphere in this region was instrumental in the slowdown and shoaling of deep-water production in the subpolar North Atlantic” is contradictory to that statement.

In our initial submission we discussed the changes in the BPWO as related to the variable influence of southern-source waters, based on the classic interpretation of benthic $\delta^{13}\text{C}$ records in the Atlantic Ocean (Curry & Oppo, 2005; Lynch-Stieglitz et al., 2007; Voelker et al., 2010; Raymo et al., 2004) and more recent publications using other deep-ocean tracers which suggest the same (Farmer et al. 2019; Lear et al. 2016).

During the review process we were asked (actually, by reviewer #3) to acknowledge that other processes could lead to reduction of BPWO in the North Atlantic. Thus, we discussed further, so we did it and included the papers about deep-water circulation changes during the LGM (studies by Blaser et al (2025), Howe et al. (2016) and Repschläger et al. (2021)). We repeat our response to the reviewer #3 here: “Although the current dataset available at Site U1314 does not allow an unambiguous identification of which water mass was responsible of decrease in BPWO, the lower Nd isotope values during MIS 24 and MIS 22 at mid-latitude sites in the north at sites U1313/607 (Kim et al. 2021; Yehudai et al., 2021) and 1058 (Tachikawa et al. 2021) and south Atlantic sites 1267 (Farmer et al. 2019) and U1479 (Hines et al. 2024) point out at a higher influence of southern source waters. We cannot discard the possibility of an active NADW during glacials at U1314 as suggested by the studies led by Howe et al. (2016), Repschläger et al. (2021) and Blaser et al. (2025) for the LGM and HS1. Whatever was the dominant deep-water mass, we suggest that the weakening of the AMOC and shoaling in the boundary between AABW and NADW as the main mechanism increasing storage of respired carbon during glacial intervals of the MPT.”

In our opinion, our speculations are supported by the data, and that level of caution is adequate, at a similar level as previous studies which studied the reduction of BPWO during the MPT (e.g. Thomas et al. 2022; Lear et al. 2016). This level of “speculation” or “cautioness” does not change the main idea of this study, which is the reduction in the BPWO in the subpolar North Atlantic. We hope that our study serves as a starting point and motivates the paleoceanography community to use additional proxies in this region to resolve this conundrum.

Comparison of U1314 with ODP1094: I do think that the comparison to Mn/Ca changes at ODP Site 1094 is quite a stretch. Suggesting that both cores have robust chronologies glosses a little bit over the fact that both records come with age uncertainties, in particular ODP 1094, where the benthic $\delta^{18}\text{O}$ record has quite some gaps because of poor carbonate preservation.

We disagree with the reviewer and provide arguments based on publications that refute his/her argumentation regarding ODP 1094.

First, we discard age model issues because both cores are aligned to the LR04 benthic stack using their benthic isotope record, even using the same benthic species. The latest age model for 1094 was developed by characterizing oxygen isotopes of benthic foraminifera (Hasenfratz et al., 2019) tuned to the LR04 global benthic stack (Lisiecki & Raymo, 2005). Actually, Hasenfratz et al (2019) state that *Cibicidoides* spp. are abundant throughout the entire record of ODP Sites 1094.

Second, based on the supplementary figure with the age model from Hasenfratz et al. (2019), it does not look like the isotope record has gaps which could increase the age uncertainty, and certainly not during the 1000-780 kyr interval. Consequently, the reviewer statement “in particular ODP 1094, where the benthic $\delta^{18}\text{O}$ record has quite some gaps because of poor carbonate preservation” is not supported by the data. Please, refer to the figure below.

The estimated uncertainty in the LR04 age model would be 4 kyr from 1–0 Ma (Lisiecki and Raymo, 2005), which is larger than the age offset between records from 1094 and U1314.

The statement in lines 289-291 are entirely unclear, and require more explanation. I concur with Reviewer #3 that the link to the southern hemisphere is premature and overly ambitious, still despite the revisions. In my view, ODP Site 1094 indicates changes in BWPO in southern-sourced bottom waters, in agreement with Farmer et al. (2019). Any link to the freshwater budget at the surface cannot easily be linked to convection or deep water formation in the Southern Ocean, as this happens through local processes near the Antarctic margin.

Please note that this view contrasts with reviewer #2, who considers that the comparison between 1094 and U1314 is adequate, and he/she even requests to speculate further.

We consider that a link between freshwater budget in the high-latitude North Atlantic and convection in the Southern Ocean is justified and supported by previous studies. For example, Starr et al. (2021) argue that the coupling between the southern escape of freshwater (using IRD accumulation) and deep-water mass perturbations (using benthic $\delta^{13}\text{C}$) across the MPT, implicate that freshening was a central component in setting overturning circulation state across the MPT. We now have added this study to the discussion to justify the link to the freshwater perturbations in the Southern Ocean (L. 410 and 460).

Structure of the manuscript: I must admit that I am a bit puzzled by the set up of the manuscript. After the introduction, in the section “Records of past BPWO during the MPT” the reader is referred to quite a lot of supplementary figures and extended data although this seems to be key data of the manuscript. This is confusing, and indicates that the manuscript is not as streamlined as it could/should be. Reviewer #1 has also mentioned difficulties to refer to the data in the figure. This might have improved in the revised manuscript, but referring mostly to Supplementary Material in the first quarter of the Results-Discussion section is counterproductive.

Yes, reviewer #1 pointed this out, and we made substantial changes on the structure of the manuscript during the review process. In this new round of revisions, the same reviewer says that the manuscript is now much better streamlined, and now there are not difficulties to refer to the data in the figure. Supplementary data are deemed to provide additional information only and specifics about the methodology. None of the information included there is key to following the arguments posed in the main text in the results. For example, in the supplementary the section “Sedimentary environments at Site U1314 and 1094” we discuss the details of the sedimentary processes of the sites that we are comparing, but this information is not crucial for following the main text.

New data: I am a bit confused what data are newly presented. In line 105, it is stated that “previously published Mn/Al ratios in bulk sediment and Mn/Ca ratio in planktonic coatings at those two sites are used as proxies for pore water oxygenation”. Here I would like to see references to the publications in the text, rather than reference to “Supplementary Information”.

References have been added to the text maintained in the supplementary information.

Some reviewers mentioned discrepancies between some interglacial stages in the response of the BPWO proxies, and inter-proxy differences. The authors have addressed these but there remains uncertainties in how valid absolute BPWO reconstructions based on these semiquantitative/qualitative proxies are.

We did not attempt stating absolute BPWO values in our study, since there is no calibration for the proxies used in this study for the study area. Even studies attempting to provide oxygen values using a specific calibration for a proxy show limitations in the upper end of the calibration (Thomas et al., 2022).

It shows in some of the proxy responses during some interglacials (not as high as expected) or terminations (still low oxygen despite onset of interglaciation). There is some uncertainty around redox-sensitive proxies that let's one doubt whether these features are true signals or in fact sedimentary artifacts (due to changes in redox-front etc.). The sections on sedimentation changes helps but there could still be changes in sediment composition, sediment porosity that drives the redox-proxies, although I believe the wide-spread similar response in BPWO in the North Atlantic supports the findings.

As mentioned in the supplementary material and in the reply to the reviewers, the overall agreement between proxies of different nature at the same site and when compared to BPWO proxies from other North Atlantic cores excludes that sedimentary artifacts are driving the downcore signal. The different magnitude of changes or delay in reoxygenation at the onset of the interglaciation is explained in L. 214.

Minor comments

We added clarity to the statements that are unclear, or answer to the reviewer's comment whenever it is necessary. The new wording is specified in red after the reviewer's comment. Many statements in the manuscript are too vague in my view, and need specification. They make it really hard to follow: "benthic foraminifera data" (l. 23), we changed it to "benthic foraminifera assemblage", "we analyzed proxies" (l. 28) we added "geochemical and micropaleontological proxies", "previously published proxies" (l. 73) we added "sedimentological" proxies, "new and previous proxy records" (l. 82) This sentence has been changed as "New and previously published BPWO proxy records"?, "previously published datasets of proxies of surface and deep oceanography" we do not feel it is unclear, since additional information about the proxies is included in the supplementary information, and listing all the proxies at both sites would make the text unnecessarily long, "Mn minerals" (l. 109) replaced by Mn-rich minerals, which can be either Mn oxides or carbonates, as described in L. 145, "geochemical data" (l. 141, I don't think IRD does not really count as geochemical data.) but that sentence refers to geochemical proxies showed in Fig. 2B-C, which correspond to geochemical proxies (P/Al, reactive P, Mn/Ca in planktonic foraminifera and Mn/Al in bulk sediments, "as the other redox proxies" (l. 155), we added "as the geochemical proxies", "in certain moments" (l. 160) we added "in some intervals during MIS 23", "as well as with other proxies indicative at Site U1314" (l. 171) this part has been rewritten and now it reads as follows "indicative of changes in surface oceanography which point to a southward advance of the AF with the consequent increase in freshwater and sea ice, as documented by other proxies at Site U1314", "using multiple independent proxies" (l. 195) we changed by "using stable isotopes and eNd", "deep-ocean circulation proxy" (l. 226; eNd is strictly a water mass provenance indicator) changed according to reviewer's suggestion; "a deep-water mass provenance indicator", "deep-water formation in convection sites" (l. 232, what convection sites are here specifically referred to? We added "deep-water formation in Greenland, Norwegian and Iceland convection sites", also in line 260), "deep-circulation records" (l. 249) we added "benthic $\delta^{13}C$ ", "lag between benthic organisms in the North Atlantic and changing deep-ocean conditions" (l. 253, how can benthic organisms lag? Do you mean their response?) we rephrased it to "lag between the response of benthic organisms in the North Atlantic and changes in deep-ocean conditions", "AABW filled the AMOC bottom cell" (l. 296, that seems to me the very definition of the lower cell) we replaced this by "the deep-ocean", "enhancing the non-linear response of the climate

system” (l. 339, what exactly is referred to here?) **we deleted that part.**

L. 48: it should say stronger interglacial ACC flow.

Added according to the reviewer’s suggestion.

L. 62: it should say “build up” or “increase in respired carbon in the subpolar North Atlantic reservoir” or similar.

Replaced by “build-up”.

L. 63: I also do not agree with the statement on the “absence of oxygen-sensitive proxies”, when studies from the area exist (Lear et al., 2016; Thomas et al., 2022).

We do not think that there is anything wrong with our statement, since sites U1385 and 607 are much downstream (37.5 °N and 41°N, respectively), far from the deep-convection areas in the northern North Atlantic, while U1314 (56°N) is the closest site with BPWO-sensitive proxies for the MPT so far.

L. 77-78: statement is very vague.

We have added ..”in setting AMOC dynamics to facilitate increased carbon sequestration”.

L. 90: AMOC lower limb, the NADW

Not sure what the suggestion is here, so we have opted for not making any changes.

L. 102: 365 $\mu\text{mol/kg}$ for AABW is too high (Figure 1 shows that [O₂] at ODP Site 1094 is much lower).

Thanks for pointing this out. Indeed the value was wrong, the correct number according to the WOA18 is 217 $\mu\text{mol/kg}$. This has been corrected accordingly.

L. 110: Add reference to sentence that ends in line 110.

Reference added, Mangini, A., A. Eisenhauer, and P. Walter, Response of Manganese in the Ocean to the Climatic Cycles in the Quaternary. *Paleoceanography*, 1990. 5(5): p. 811–821.

L. 126: Add reference to sentence that ends in line 126.

Reference added: Van Cappellen, Philippe, and Ellery D. Ingall. "Benthic phosphorus regeneration, net primary production, and ocean anoxia: A model of the coupled marine biogeochemical cycles of carbon and phosphorus." *Paleoceanography* 9.5 (1994): 677-692.

L. 136-138: weird phrasing (“we attribute changes in BPWO [...] to changes in deep-ocean oxygenation”). I see that there the physical/dynamical driver of oxygen is intended to be emphasized, but this should be specified.

We agree with the reviewer and we have replaced “oxygenation” by “circulation”.

L. 145. Add reference or figure callout to sentence ending in line 145

We added a callout to Fig. 2D

L. 158. Add reference or figure callout to sentence ending in line 158

We added a callout to Fig. 2F.

L. 169. Add reference or figure callout to sentence ending in line 169

We added a callout to Fig. 2B-D.

L. 187. The reason is unclear why specifically “Rockall Plateau” is important here, which is quite a narrow area in the North Atlantic.

We referred specifically to the Rockall Plateau, as well as Iceland and Irminger basin, because these are the regions that the references included in that sentence referred to.

L. 188. Irminger Basin

Thanks for spotting the typo, corrected.

L. 216: 5-15 $\mu\text{mol/kg}$? I also recommend to mention where IODP Site U1385 is located.

Numbers reordered, and added SW Iberian Margin.

L. 218+222+263+280+282. Add reference and/or figure callout to sentence ending in line 218+222+263+280+282

Callouts to figures added for L. 218, 222, 263. However, L. 280 and 282 have figure callouts, so we do not consider additional information is needed.

L. 221. Define Dd13Cb

This has been defined in L. 221 of the revised manuscript, and used instead $\Delta\delta^{13}\text{C}_{\text{wue-aff}}$ to agree with the definition of this proxy in the caption of figure 2.

L. 226. I don't see how eNd can information about the AMOC strength, when this proxy mostly indicates water mass origin

This has been rephrased and now it reads as follows: “deep-water mass provenance indicator”. Please note that specifically eNd is referred as “deep ocean circulation proxy” by Hines et al 2025.

L. 240. “Higher ice mass” (as in tall) or more expanded ice sheets (as in more area)? Please

specify.

We changed it to “larger” and “expanded” (L. 326 and 324).

L. 251 and figure axes labels: “well ventilated benthic foraminiferal assemblages” is a strange phrasing. It is not the foraminifera that are well ventilated, but the bottom water environment that they track. Please rephrase.

Replaced by “benthic foraminifera assemblage related to well-ventilated waters”

L. 301. Unclear why a number of eNd studies are cited when referring to “increasing trends in deep ocean nutrient and dissolved inorganic carbon concentrations”

Yehudai, Tachikawa, Kim and Jaume-Seguí papers have been removed from that sentence.

L. 425. “Foraminiferal identifications were based on specialized literature”. Either include citations or remove.

Rephrased, now it reads as follows “Foraminiferal identifications follow the taxonomic concepts of WoRMS (last accessed on 2024-05-28) [98]”.

I wonder why the authors haven’t applied more quantitative bottom water oxygen assessments based on the BFOI index of benthic foraminiferal assemblages or similar.

Thanks for the suggestion. Unfortunately, several of the species dominant at U1314 (e.g. *Astrononion novozealandicum* and *Gyroidina umbonata*) are not included in the list of species for the BFOI index, so it is not possible to make any BPWO assessment based on it.

References

Andrews, John T. "Icebergs and iceberg rafted detritus (IRD) in the North Atlantic: facts and assumptions." *Oceanography* (2000): 100-108.

Barker, S. et al. Icebergs not the trigger for North Atlantic cold events. *Nature*, 520 (2015), pp. 333-336, 10.1038/nature14330

Broecker, Wallace S. "Massive iceberg discharges as triggers for global climate change." *Nature* 372.6505 (1994): 421-424.

Herbert, Timothy D. "The mid-Pleistocene climate transition." *Annual Review of Earth and Planetary Sciences* 51.1 (2023): 389-418.

Hönisch, Bärbel, et al. "Atmospheric carbon dioxide concentration across the mid-Pleistocene transition." *Science* 324.5934 (2009): 1551-1554.

Bjørklund, K. R., and P. F. Ciesielski. "Ecology, morphology, stratigraphy, and the paleoceanographic significance of *Cycladophora davisiana davisiana*. Part I: Ecology and morphology." *Marine Micropaleontology* 24.1 (1994): 71-88.

Calvert, S. E., and T. F. Pedersen. "Chapter fourteen elemental proxies for palaeoclimatic and palaeoceanographic variability in marine sediments: interpretation and application." *Developments in marine geology* 1 (2007): 567-644.

Carstens, Jörn, and Gerold Wefer. "Recent distribution of planktonic foraminifera in the Nansen Basin, Arctic Ocean." *Deep Sea Research Part A. Oceanographic Research Papers* 39.2 (1992): S507-S524.

Chalk, Thomas B., et al. "Causes of ice age intensification across the Mid-Pleistocene Transition." *Proceedings of the National Academy of Sciences* 114.50 (2017): 13114-13119.

Clark, Peter U., et al. "The middle Pleistocene transition: characteristics, mechanisms, and implications for long-term changes in atmospheric pCO₂." *Quaternary Science Reviews* 25.23-24 (2006): 3150-3184.

Curry, W.B., Oppo, D.W., 2005. Glacial water mass geometry and the distribution of $\delta^{13}\text{C}$ of ΣCO_2 in the western Atlantic Ocean. *Paleoceanography* 20, PA1017,doi: 10.1029/2004PA001021.

Elderfield, Henry, et al. "Evolution of ocean temperature and ice volume through the mid-Pleistocene climate transition." *science* 337.6095 (2012): 704-709.

Farmer, J., et al., Deep Atlantic Ocean carbon storage and the rise of 100,000-year glacial cycles. *Nature Geoscience*, 2019. 12(5): p. 355-360

Gruetzner, Jens, and S. M. Higgins. "Threshold behavior of millennial scale variability in deep water hydrography inferred from a 1.1 Ma long record of sediment provenance at the southern Gardar Drift." *Paleoceanography* 25.4 (2010).

Hines, Sophia KV, et al. "Revisiting the mid-Pleistocene transition ocean circulation crisis." *Science* 386.6722 (2024): 681-686.

Kretschmer, Kerstin, Michal Kucera, and Michael Schulz. "Modeling the distribution and seasonality of *Neogloboquadrina pachyderma* in the North Atlantic Ocean during Heinrich Stadial 1." *Paleoceanography* 31.7 (2016): 986-1010. Lear, C.H., et al., Breathing more deeply: Deep ocean carbon storage during the mid-Pleistocene climate transition. *Geology*, 2016. 44(12): p. 1035-1038.

Lisiecki, Lorraine E., and Maureen E. Raymo. "A Pliocene-Pleistocene stack of 57 globally distributed benthic $\delta^{18}\text{O}$ records." *Paleoceanography* 20.1 (2005).

Lynch-Stieglitz, Jean, et al. "Atlantic meridional overturning circulation during the Last Glacial Maximum." *science* 316.5821 (2007): 66-69.

McManus, J., Oppo, D.W., Cullen, J.L., 1999. A 0.5-million-year record of millennial-scale climate variability in the North Atlantic. *Science* 283, 971–975

Morley, Joseph J., and James D. Hays. "Oceanographic conditions associated with high abundances of the radiolarian *Cycladophora davisiana*." *Earth and Planetary Science Letters* 66 (1983): 63-72.

Muglia, Juan, and Andreas Schmittner. "Carbon isotope constraints on glacial Atlantic meridional overturning: Strength vs depth." *Quaternary Science Reviews* 257 (2021): 106844.

Okazaki, Yusuke, et al. "The production scheme of *Cycladophora davisiana* (Radiolaria) in the Okhotsk Sea and the northwestern North Pacific: implication for the paleoceanographic conditions during the glacials in the high latitude oceans." *Geophysical Research Letters* 30.18 (2003).

Parnell-Turner, R., Briais, A., LeVay, L., Morris, M., Cui, Y., Di Chiara, A., Dodd, J. P., Dunkley Jones, T., Dwyer, D., Eason, D., Friedman, S., Hemming, S., Hochmuth, K., Ibrahim, H. E., Jasper, C., Karatsolis, B.-T., Lee, S., LeBlanc, D., Lindsay, M., ... Sperling, C. (2025).

IODP Expedition 395 Site U1602 X-ray fluorescence (XRF) (Version v1) [Data set]. International Ocean Discovery Program. <https://doi.org/10.5281/zenodo.15065344>

Pena, Leopoldo D., and Steven L. Goldstein. "Thermohaline circulation crisis and impacts during the mid-Pleistocene transition." *Science* 345.6194 (2014): 318-322.

Raymo, Maureen E., et al. "Stability of North Atlantic water masses in face of pronounced climate variability during the Pleistocene." *Paleoceanography* 19.2 (2004).

Sinnesael, Matthias, et al. "Onset of strong Iceland-Scotland overflow water 3.6 million years ago." *Nature communications* 16.1 (2025): 43

Starr, Aidan, et al. "Antarctic icebergs reorganize ocean circulation during Pleistocene glacials." *Nature* 589.7841 (2021): 236-241.

Stevenard, N., Kissel, C., Govin, A., Wandres, C., 2024. Deep north atlantic circulation strength: Glacial-interglacial variability over the last 400,000 years. *Quaternary Science Reviews* 345, 109011, doi: <https://doi.org/10.1016/j.quascirev.2024.109011>

Thomas, N.C., H.J. Bradbury, and D.A. Hodell, Changes in North Atlantic deep-water oxygenation across the Middle Pleistocene Transition. *Science*, 2022. 377(6606): p. 654-659.

Venz, K.A., Hodell, D.A., Stanton, C., Warnke, D.A., 1999. A 1.0 Myr record of Glacial North Atlantic Intermediate Water variability from ODP site 982 in the northeast Atlantic. *Paleoceanography* 14, 42–52,

Voelker, Antje HL, et al. "Variations in mid-latitude North Atlantic surface water properties during the mid-Brunhes (MIS 9–14) and their implications for the thermohaline circulation." *Climate of the Past* 6.4 (2010): 531-552.

Weitkamp, T.M., Bird, C., Darling, K.F., Hsiang, A.Y., Ramsay, J., Vermassen, F., Coxall, H.K., 2025. Aberrant coiling signatures reveal the specialised reproductive strategy of the planktonic foraminifera *Neogloboquadrina pachyderma* under Central Arctic perennial sea ice. *Marine Micropaleontology* 201, 102503, doi: <https://doi.org/10.1016/j.marmicro.2025.102503>

Wright, Amy K., and Benjamin P. Flower. "Surface and deep ocean circulation in the subpolar North Atlantic during the mid-Pleistocene revolution." *Paleoceanography* 17.4 (2002): 20-1.